# ENERGY-BASED CONCEPT BOTTLENECK MODELS: UNIFYING PREDICTION, CONCEPT INTERVENTION, AND PROBABILISTIC INTERPRETATIONS

**Xinyue Xu[1], Yi Qin[1], Lu Mi[2], Hao Wang[3†], Xiaomeng Li[1†]**
[1]The Hong Kong University of Science and Technology, [2]University of Washington,
[3]Rutgers University, [†]Equal advising
{xxucb, yqinar, eexmli}@ust.hk, milu@uw.edu, hw488@cs.rutgers.edu

## ABSTRACT

Existing methods, such as concept bottleneck models (CBMs), have been successful in providing concept-based interpretations for black-box deep learning models. They typically work by predicting concepts given the input and then predicting the final class label given the predicted concepts. However, (1) they often fail to capture the high-order, nonlinear interaction between concepts, e.g., correcting a predicted concept (e.g., "yellow breast") does not help correct highly correlated concepts (e.g., "yellow belly"), leading to suboptimal final accuracy; (2) they cannot naturally quantify the complex conditional dependencies between different concepts and class labels (e.g., for an image with the class label "Kentucky Warbler" and a concept "black bill", what is the probability that the model correctly predicts another concept "black crown"), therefore failing to provide deeper insight into how a black-box model works. In response to these limitations, we propose **Energy-based Concept Bottleneck Models (ECBMs)**. Our ECBMs use a set of neural networks to define the joint energy of candidate (input, concept, class) tuples. With such a unified interface, prediction, concept correction, and conditional dependency quantification are then represented as conditional probabilities, which are generated by composing different energy functions. Our ECBMs address both limitations of existing CBMs, providing higher accuracy and richer concept interpretations. Empirical results show that our approach outperforms the state-of-the-art on real-world datasets.

## 1 INTRODUCTION

Black-box models, while powerful, are often unable to explain their predictions in a way that is comprehensible to humans (Rudin, 2019). Concept-based models aim to address this limitation. Unlike traditional end-to-end models (Zhang et al., 2021) predicting output directly from input, concept-based models first predict intermediate concepts from input and then predict the final class labels from the predicted concepts (Koh et al., 2020; Kazhdan et al., 2020). These models aim to emulate humans' cognitive process of distinguishing between different objects (e.g., zoologists classifying birds according to their heads, wings, and tails) by generating concepts that are visually comprehensible to humans as intermediate interpretations for their predictions.

Concept Bottleneck Models (CBMs) (Koh et al., 2020), as a representative class of models, operate by firstly generating concepts given the input and then using these concepts to predict the final label. The vanilla CBMs often fall short in final prediction accuracy compared to black-box models, creating a potentially unnecessary performance-interpretability trade-off (Rudin et al., 2022). To improve on such trade-off, Concept Embedding Models (CEMs) (Zarlenga et al., 2022) improve CBMs by including positive and negative semantics, while Post-hoc Concept Bottleneck Models (PCBMs) (Yuksekgonul et al., 2022) make use of residual fitting to compensate for limitations in concept learning. Despite recent advances, existing CBM variants (including CEMs and PCBMs) still suffer from the following key limitations:

1. **Interpretability:** They cannot effectively quantify the intricate relationships between various concepts and class labels (for example, in an image labeled "Kentucky Warbler", what is the likelihood that the model accurately identifies the concept "black crown"). As a result, they fall short of offering deeper understanding into the workings of a black-box model.
2. **Intervention:** They often struggle to account for the complex interactions among concepts. Consequently, intervening to correct a misidentified concept (e.g., "yellow breast") does not necessarily improve the accuracy of closely related concepts (e.g., "yellow belly"). This limitation results in suboptimal accuracy for both individual concepts and the final class label.
3. **Performance:** Current CBM variants suffer from a trade-off (Zarlenga et al., 2022) between model performance and interpretability. However, an ideal interpretable model should harness the synergy between performance and interpretability to get the best of both worlds.

In response to these limitations, we propose **Energy-based Concept Bottleneck Models (ECBMs)**. Our ECBMs use a set of neural networks to define the joint energy of the input $x$, concept $c$, and class label $y$. With such a unified interface, (1) prediction of the class label $y$, (2) prediction of concepts $c_{-k}$ (i.e., all concepts except for $c_k$) after correcting concept $c_k$ for input $x$, and (3) conditional interpretation among class label $y$, concept $c_k$, and another concept $c_{k'}$ can all be naturally represented as conditional probabilities $p(y|x)$, $p(c_{-k}|x, c_k)$, and $p(c_k|y, c_{k'})$, respectively; these probabilities are then easily computed by composing different energy functions.

We summarize our contributions as follows:

- Beyond typical concept-based prediction, we identify the problems of concept correction and conditional interpretation as valuable tools to provide concept-based interpretations.
- We propose Energy-based Concept Bottleneck Models (ECBMs), the first general method to unify concept-based prediction, concept correction, and conditional interpretation as conditional probabilities under a joint energy formulation.
- With ECBM's unified interface, we derive a set of algorithms to compute different conditional probabilities by composing different energy functions.
- Empirical results show that our ECBMs significantly outperform the state-of-the-art on real-world datasets. Code is available at https://github.com/xmed-lab/ECBM.

## 2 RELATED WORK

**Concept Bottleneck Models** (CBMs) (Koh et al., 2020; Kumar et al., 2009; Lampert et al., 2009) use a feature extractor and a concept predictor to generate the "bottleneck" concepts, which are fed into a predictor to predict the final class labels. Concept Embedding Models (CEMs) (Zarlenga et al., 2022) build on CBMs to characterize each concept through a pair of positive and negative concept embeddings. Post-hoc Concept Bottleneck Models (PCBMs) (Yuksekgonul et al., 2022) use a post-hoc explanation model with additional residual fitting to further improve final accuracy. Probabilistic Concept Bottleneck Models (ProbCBMs) (Kim et al., 2023) incorporate probabilistic embeddings to enable uncertainty estimation of concept prediction. There are a diverse set of CBM variants (Barbiero et al., 2023; 2022; Havasi et al., 2022; Ghosh et al., 2023a;b; Yang et al., 2023; Sarkar et al., 2022; Oikarinen et al., 2023), each addressing problems from their unique perspectives. This diversity underscores the vitality of research within this field.

Here we note several key differences between the methods above and our ECBMs. (1) These approaches are inadequate at accounting for the complex, nonlinear interplay among concepts. For example, correcting a mispredicted concept does not necessarily improve the accuracy of related concepts, leading suboptimal final accuracy. (2) They cannot effectively quantify the complex conditional dependencies (detailed explanations in Appendix C.4) between different concepts and class labels, therefore failing to offer conditional interpretation on how a black-box model works. In contrast, our ECBMs address these limitations by defining the joint energy of candidate (input, concept, class) tuples and unify both concept correction and conditional interpretation as conditional probabilities, which are generated by composing different energy functions.

**Energy-Based Models** (LeCun et al., 2006; Tu et al., 2020; Deng et al., 2020; Nijkamp et al., 2020) leverage Boltzmann distributions to decide the likelihood of input samples, mapping each sample to a scalar energy value through an energy function. The development of energy-based models have been signficantly influenced by pioneering works such as (Xie et al., 2016) and (Xie et al., 2018).

Beyond classification (Li et al., 2022; Grathwohl et al., 2019), energy-based models have also been applied to structured prediction tasks (Belanger & McCallum, 2016; Rooshenas et al., 2019; Tu & Gimpel, 2019). Xie et al. and Du et al. use energy-based models for the distribution of data and labels, which also capture concepts. These methods use energy functions to improve prediction performance, but cannot provide concept-based interpretations. In contrast, our ECBMs estimate the joint energy of input, concepts, and class labels, thereby naturally providing comprehensive concept-based interpretations that align well with human intuition.

**Unsupervised Concept-Based Models**, unlike CBMs, aim to extract concepts without concept annotations. This is achieved by introducing inductive bias based on Bayesian deep learning with probabilistic graphical models (Wang et al., 2019; Wang & Yeung, 2016; 2020; Wang & Yan, 2023; Xu et al., 2023), causal structure (Lin et al., 2022), clustering structure (Chen et al., 2019; Ma et al., 2023), generative models (Du et al., 2021; Liu et al., 2023a) or interpretability desiderata (Alvarez Melis & Jaakkola, 2018).

## 3 ENERGY-BASED CONCEPT BOTTLENECK MODELS

In this section, we introduce the notation, problem settings, and then our proposed ECBMs in detail.

**Notation.** We consider a supervised classification setting with $N$ data points, $K$ concepts, and $M$ classes, namely $\mathcal{D} = (\boldsymbol{x}^{(j)}, \boldsymbol{c}^{(j)}, \boldsymbol{y}^{(j)})_{j=1}^{N}$, where the $j$-th data point consists of the input $\boldsymbol{x}^{(j)} \in \mathcal{X}$, the label $\boldsymbol{y}^{(j)} \in \mathcal{Y} \subset \{0,1\}^{M}$, and the concept $\boldsymbol{c}^{(j)} \in \mathcal{C} = \{0,1\}^{K}$; note that $\mathcal{Y}$ is the space of $M$-dimensional one-hot vectors while $\mathcal{C}$ is not. We denote as $\boldsymbol{y}_{m} \in \mathcal{Y}$ the $M$-dimensional one-hot vector with the $m$-th dimension set to 1, where $m \in \{1, \ldots, M\}$. $c_{k}^{(j)}$ denotes the $k$-th dimension of the concept vector $\boldsymbol{c}^{(j)}$, where $k \in \{1, \ldots, K\}$. We denote $[c_{i}^{(j)}]_{i \neq k}$ as $\boldsymbol{c}_{-k}^{(j)}$ for brevity. A pretrained backbone neural network $F : \mathcal{X} \rightarrow \mathcal{Z}$ is used to extract the features $\boldsymbol{z} \in \mathcal{Z}$ from the input $\boldsymbol{x} \in \mathcal{X}$. Finally, the structured energy network $E_{\boldsymbol{\theta}}(\cdot, \cdot)$ parameterized by $\boldsymbol{\theta}$, maps the $(\boldsymbol{x}, \boldsymbol{y})$, $(\boldsymbol{x}, \boldsymbol{c})$, or $(\boldsymbol{c}, \boldsymbol{y})$ to real-valued scalar energy values. We omit the superscript $^{(j)}$ when the context is clear.

**Problem Settings.** For each data point, we consider three problem settings:

1. **Prediction ($p(\boldsymbol{c}, \boldsymbol{y}|\boldsymbol{x})$).** This is the typical setting for concept-based models; given the input $\boldsymbol{x}$, the goal is to predict the class label $\boldsymbol{y}$ and the associated concepts $\boldsymbol{c}$ to interpret the predicted class label. Note that CBMs decompose $p(\boldsymbol{c}, \boldsymbol{y}|\boldsymbol{x})$ to predict $p(\boldsymbol{c}|\boldsymbol{x})$ and then $p(\boldsymbol{y}|\boldsymbol{c})$.
2. **Concept Correction/Intervention (e.g., $p(\boldsymbol{c}_{-k}|\boldsymbol{x}, c_{k})$).** Given the input $\boldsymbol{x}$ and a corrected concept $c_{k}$, predict all the other concepts $\boldsymbol{c}_{-k}$.
3. **Conditional Interpretations (Wang et al., 2019) (e.g., $p(\boldsymbol{c}|\boldsymbol{y})$ or $p(c_{k}|\boldsymbol{y}, c_{k'})$).** Interpret the model using *conditional probabilities* such as $p(c_{k}|\boldsymbol{y}, c_{k'})$ (i.e., given an image with class label $\boldsymbol{y}$ and concept $c_{k'}$, what is the probability that the model correctly predicts concept $c_{k}$).

### 3.1 STRUCTURED ENERGY-BASED CONCEPT BOTTLENECK MODELS

**Overview.** Our ECBM consists of three energy networks collectively parameterized by $\boldsymbol{\theta}$: (1) a class energy network $E_{\boldsymbol{\theta}}^{class}(\boldsymbol{x}, \boldsymbol{y})$ that measures the compatibility of input $\boldsymbol{x}$ and class label $\boldsymbol{y}$, (2) a concept energy network $E_{\boldsymbol{\theta}}^{concept}(\boldsymbol{x}, \boldsymbol{c})$ that measures the compatibility of input $\boldsymbol{x}$ and the $K$ concepts $\boldsymbol{c}$, and (3) a global energy network $E_{\boldsymbol{\theta}}^{global}(\boldsymbol{c}, \boldsymbol{y})$ that measures the compatability of the $K$ concepts $\boldsymbol{c}$ and class label $\boldsymbol{y}$. The *class* and *concept* energy networks model *class labels* and *concepts* separately; in contrast, the *global* energy network model the *global relation* between class labels and concepts. For all three energy networks, *lower energy* indicates *better compatibility*. ECBM is trained by minimizing the following total loss function:

$$\mathcal{L}_{total}^{all} = \mathbb{E}_{(\boldsymbol{x}, \boldsymbol{c}, \boldsymbol{y}) \sim p_{\mathcal{D}}(\boldsymbol{x}, \boldsymbol{c}, \boldsymbol{y})}[\mathcal{L}_{total}(\boldsymbol{x}, \boldsymbol{c}, \boldsymbol{y})] \tag{1}$$

$$\mathcal{L}_{total}(\boldsymbol{x}, \boldsymbol{c}, \boldsymbol{y}) = \mathcal{L}_{class}(\boldsymbol{x}, \boldsymbol{y}) + \lambda_{c}\mathcal{L}_{concept}(\boldsymbol{x}, \boldsymbol{c}) + \lambda_{g}\mathcal{L}_{global}(\boldsymbol{c}, \boldsymbol{y}), \tag{2}$$

where $\mathcal{L}_{class}$, $\mathcal{L}_{concept}$, and $\mathcal{L}_{global}$ denote the loss for training the three energy networks $E_{\boldsymbol{\theta}}^{class}(\boldsymbol{x}, \boldsymbol{y})$, $E_{\boldsymbol{\theta}}^{concept}(\boldsymbol{x}, \boldsymbol{c})$, and $E_{\boldsymbol{\theta}}^{global}(\boldsymbol{c}, \boldsymbol{y})$, respectively. $\lambda_{c}$ and $\lambda_{g}$ are hyperparameters. Fig. 1 shows an overview of our ECBM. Below we discuss the three loss terms (Eqn. 1) in detail.

**Class Energy Network $E_{\boldsymbol{\theta}}^{class}(\boldsymbol{x}, \boldsymbol{y})$.** In our ECBM, each class $m$ is associated with a trainable class embedding denoted as $\boldsymbol{u}_{m}$. As shown in Fig. 1(top), given the input $\boldsymbol{x}$ and a candidate label

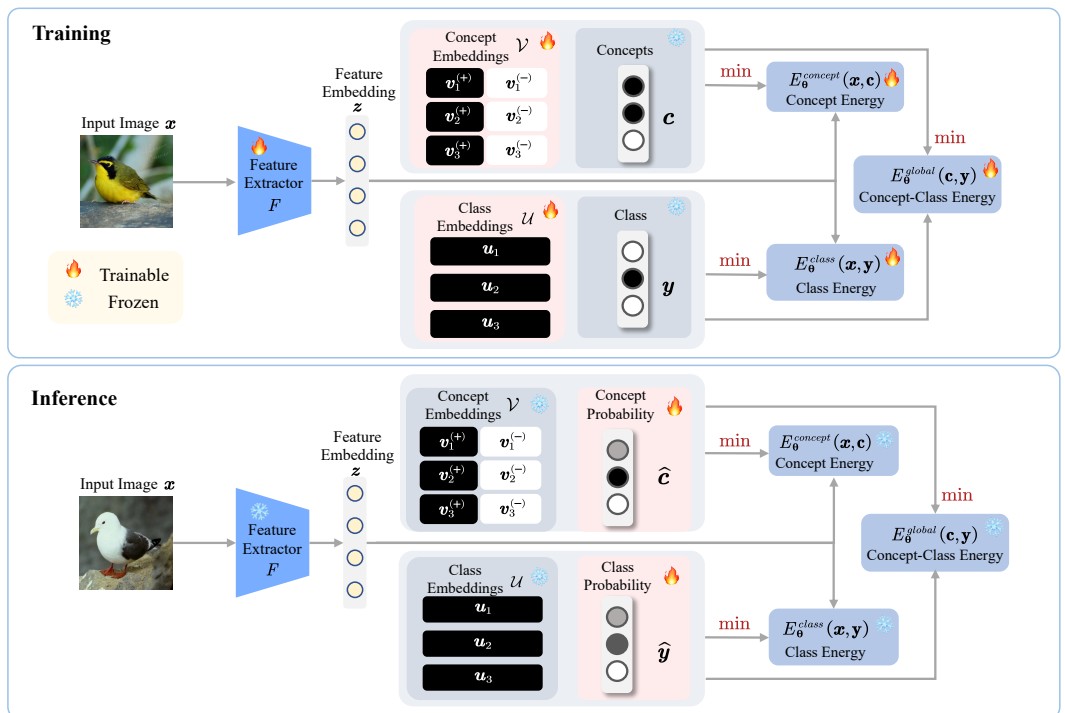

Figure 1: Overview of our ECBM. **Top:** During training, ECBM learns positive concept embeddings $\boldsymbol{v}_k^{(+)}$ (in black), negative concept embeddings $\boldsymbol{v}_k^{(-)}$ (in white), the class embeddings $\boldsymbol{u}_m$ (in black), and the three energy networks by minimizing the three energy functions, $E_{\boldsymbol{\theta}}^{class}(\boldsymbol{x}, \boldsymbol{y})$, $E_{\boldsymbol{\theta}}^{concept}(\boldsymbol{x}, \boldsymbol{c})$, and $E_{\boldsymbol{\theta}}^{global}(\boldsymbol{c}, \boldsymbol{y})$ using Eqn. 1. The concept $\boldsymbol{c}$ and class label $\boldsymbol{y}$ are treated as constants. **Bottom:** During inference, we (1) freeze all concept and class embeddings as well as all networks, and (2) update the predicted concept probabilities $\widehat{\boldsymbol{c}}$ and class probabilities $\widehat{\boldsymbol{y}}$ by minimizing the three energy functions using Eqn. 1.

$\boldsymbol{y}$, the feature extractor $F$ first compute the features $\boldsymbol{z} = F(\boldsymbol{x})$. We then feed $\boldsymbol{y}$'s associated class label embedding $\boldsymbol{u}$ along with the features $\boldsymbol{z}$ into a neural network $G_{zu}(\boldsymbol{z}, \boldsymbol{u})$ to obtain the final $E_{\boldsymbol{\theta}}^{class}(\boldsymbol{x}, \boldsymbol{y})$. Formmaly we have,

$$E_{\boldsymbol{\theta}}^{class}(\boldsymbol{x}, \boldsymbol{y}) = G_{zu}(\boldsymbol{z}, \boldsymbol{u}), \tag{3}$$

where $G_{zu}(\cdot, \cdot)$ is a trainable neural network. To train the class energy network, we use the Boltzmann distribution to define the conditional likelihood of $\boldsymbol{y}$ given input $\boldsymbol{x}$:

$$p_{\boldsymbol{\theta}}(\boldsymbol{y}|\boldsymbol{x}) = \frac{\exp\left(-E_{\boldsymbol{\theta}}^{class}(\boldsymbol{x}, \boldsymbol{y})\right)}{\sum_{m=1}^{M} \exp\left(-E_{\boldsymbol{\theta}}^{class}(\boldsymbol{x}, \boldsymbol{y}_m)\right)}, \tag{4}$$

where the denominator serves as a normalizing constant. $\boldsymbol{y}_m \in \mathcal{Y}$ a one-hot vector with the $m$-th dimension set to 1. The class energy network $E_{\boldsymbol{\theta}}^{class}(\boldsymbol{x}, \boldsymbol{y})$ is parameterized by $\boldsymbol{\theta}$; it maps the input-class pair $(\boldsymbol{x}, \boldsymbol{y})$ to a real-valued scalar energy. Our ECBM uses the negative log-likelihood as the loss function; for an input-class pair $(\boldsymbol{x}, \boldsymbol{y})$:

$$\mathcal{L}_{class}(\boldsymbol{x}, \boldsymbol{y}) = -\log p_{\boldsymbol{\theta}}(\boldsymbol{y}|\boldsymbol{x}) = E_{\boldsymbol{\theta}}^{class}(\boldsymbol{x}, \boldsymbol{y}) + \log\left(\sum_{m=1}^{M} e^{-E_{\boldsymbol{\theta}}^{class}(\boldsymbol{x}, \boldsymbol{y}_m)}\right). \tag{5}$$

**Concept Energy Network $E_{\boldsymbol{\theta}}^{concept}(\boldsymbol{x}, \boldsymbol{c})$.** Our concept energy network $E_{\boldsymbol{\theta}}^{concept}(\boldsymbol{x}, \boldsymbol{c})$ consists of $K$ sub-networks, $E_{\boldsymbol{\theta}}^{concept}(\boldsymbol{x}, c_k)$ where $k \in \{1, \dots, K\}$. Each sub-network $E_{\boldsymbol{\theta}}^{concept}(\boldsymbol{x}, c_k)$ measures the compatibility of the input $\boldsymbol{x}$ and the $k$-th concept $c_k \in \{0, 1\}$. Each concept $k$ is associated with a positive embedding $\boldsymbol{v}_k^{(+)}$ and a negative embedding $\boldsymbol{v}_k^{(-)}$. We define the $k$-th concept embedding $\boldsymbol{v}_k$ as a combination of positive and negative embeddings, weighted by the concept probability $c_k$, i.e., $\boldsymbol{v}_k = c_k \cdot \boldsymbol{v}_k^{(+)} + (1 - c_k) \cdot \boldsymbol{v}_k^{(-)}$. As shown in Fig. 1(top), given the input $\boldsymbol{x}$ and an concept $c_k$, the feature extractor $F$ first compute the features $\boldsymbol{z} = F(\boldsymbol{x})$. We then feed $c_k$'s associated concept embedding $(\boldsymbol{v}_k^{(+)})$ if $c_k = 1$ and $(\boldsymbol{v}_k^{(-)})$ if $c_k = 0$ along with the features $\boldsymbol{z}$ into

a neural network to obtain the final $E_{\boldsymbol{\theta}}^{concept}(\boldsymbol{x}, c_k)$. Formally, we have

$$E_{\boldsymbol{\theta}}^{concept}(\boldsymbol{x}, c_k) = G_{zv}(\boldsymbol{z}, \boldsymbol{v}_k), \tag{6}$$

where $G_{zv}(\cdot, \cdot)$ is a trainable neural network. Similar to the class energy network (Eqn. 5), the loss function for training the $k$-th sub-network $E_{\boldsymbol{\theta}}^{concept}(\boldsymbol{x}, c_k)$ is

$$\mathcal{L}_{concept}^{(k)}(\boldsymbol{x}, c_k) = E_{\boldsymbol{\theta}}^{concept}(\boldsymbol{x}, c_k) + \log\left(\sum_{c_k \in \{0,1\}} e^{-E_{\boldsymbol{\theta}}^{concept}(\boldsymbol{x}, c_k)}\right). \tag{7}$$

Therefore, for each input-concept pair $(\boldsymbol{x}, \boldsymbol{c})$, the loss function for training $E_{\boldsymbol{\theta}}^{concept}(\boldsymbol{x}, \boldsymbol{c})$ is

$$\mathcal{L}_{concept}(\boldsymbol{x}, \boldsymbol{c}) = \sum_{k=1}^{K} \mathcal{L}_{concept}^{(k)}(\boldsymbol{x}, c_k). \tag{8}$$

**Global Energy Network $E_{\boldsymbol{\theta}}^{global}(\boldsymbol{c}, \boldsymbol{y})$.** The class energy network learns the dependency between the input and the class label, while the concept energy network learns the dependency between the input and each concept separately. In contrast, our global energy network learns (1) the interaction between different concepts and (2) the interaction between all concepts and the class label.

Given the class label $\boldsymbol{y}$ and the concepts $\boldsymbol{c} = [c_k]_{k=1}^{K}$, we will feed $\boldsymbol{y}$'s associated class label embedding $\boldsymbol{u}$ along with $\boldsymbol{c}$'s associated $K$ concept embeddings $[\boldsymbol{v}_k]_{k=1}^{K}$ ($\boldsymbol{v}_k = \boldsymbol{v}_k^{(+)}$) if $c_k = 1$ and ($\boldsymbol{v}_k = \boldsymbol{v}_k^{(-)}$) if $c_k = 0$ into a neural network to compute the global energy $E_{\boldsymbol{\theta}}^{global}(\boldsymbol{c}, \boldsymbol{y})$. Formally, we have

$$E_{\boldsymbol{\theta}}^{global}(\boldsymbol{c}, \boldsymbol{y}) = G_{vu}([\boldsymbol{v}_k]_{k=1}^{K}, \boldsymbol{u}), \tag{9}$$

where $G_{vu}(\cdot, \cdot)$ is a trainable neural network. $[\boldsymbol{v}_k]_{k=1}^{k}$ denotes the concatenation of all concept embeddings. For each concept-class pair $(\boldsymbol{c}, \boldsymbol{y})$, the loss function for training $E_{\boldsymbol{\theta}}^{global}(\boldsymbol{c}, \boldsymbol{y})$ is

$$\mathcal{L}_{global}(\boldsymbol{c}, \boldsymbol{y}) = E_{\boldsymbol{\theta}}^{global}(\boldsymbol{c}, \boldsymbol{y}) + \log\left(\sum_{m=1, \boldsymbol{c}' \in \mathcal{C}}^{M} e^{-E_{\boldsymbol{\theta}}^{global}(\boldsymbol{c}', \boldsymbol{y}_m)}\right), \tag{10}$$

where $\boldsymbol{c}'$ enumerates all concept combinations in the space $\mathcal{C}$. In practice, we employ a negative sampling strategy to enumerate a subset of possible combinations for computational efficiency.

**Inference Phase.** After training ECBM using Eqn. 1, we can obtain the feature extractor $F$ and energy network parameters $\boldsymbol{\theta}$ (including class embeddings $[\boldsymbol{u}_m]_{m=1}^{M}$, concept embeddings $[\boldsymbol{v}_k]_{k=1}^{K}$, as well as the parameters of neural networks $G_{zu}(\cdot, \cdot)$, $G_{zv}(\cdot, \cdot)$, and $G_{vu}(\cdot, \cdot)$). During inference, we will freeze all parameters $F$ and $\boldsymbol{\theta}$ to perform (1) prediction of concepts and class labels (Sec. 3.2), (2) concept correction/intervention (Sec. 3.3), and (3) conditional interpretations (Sec. 3.4). Below we provide details on these three inference problems.

## 3.2 PREDICTION

To predict $\boldsymbol{c}$ and $\boldsymbol{y}$ given the input $\boldsymbol{x}$, we freeze the feature extractor $F$ and the energy network parameters $\boldsymbol{\theta}$ and search for the optimal prediction of concepts $\widehat{\boldsymbol{c}}$ and the class label $\widehat{\boldsymbol{y}}$ as follows:

$$\arg\min_{\widehat{\boldsymbol{c}}, \widehat{\boldsymbol{y}}} \quad \mathcal{L}_{class}(\boldsymbol{x}, \widehat{\boldsymbol{y}}) + \lambda_c \mathcal{L}_{concept}(\boldsymbol{x}, \widehat{\boldsymbol{c}}) + \lambda_g \mathcal{L}_{global}(\widehat{\boldsymbol{c}}, \widehat{\boldsymbol{y}}), \tag{11}$$

where $\mathcal{L}_{class}(\cdot, \cdot)$, $\mathcal{L}_{concept}(\cdot, \cdot)$, and $\mathcal{L}_{global}(\cdot, \cdot)$ are the instance-level loss functions in Eqn. 5, Eqn. 8, and Eqn. 10, respectively. Since the second term of these three loss functions remain constant during inference, one only needs to minimize the joint energy below:

$$E_{\boldsymbol{\theta}}^{joint}(\boldsymbol{x}, \boldsymbol{c}, \boldsymbol{y}) \triangleq E_{\boldsymbol{\theta}}^{class}(\boldsymbol{x}, \boldsymbol{y}) + \lambda_c E_{\boldsymbol{\theta}}^{concept}(\boldsymbol{x}, \boldsymbol{c}) + \lambda_g E_{\boldsymbol{\theta}}^{global}(\boldsymbol{c}, \boldsymbol{y}). \tag{12}$$

Therefore Eqn. 11 is simplified to $\arg\min_{\widehat{\boldsymbol{c}}, \widehat{\boldsymbol{y}}} E_{\boldsymbol{\theta}}^{joint}(\boldsymbol{x}, \widehat{\boldsymbol{c}}, \widehat{\boldsymbol{y}})$. To make the optimization tractable, we relax the support of $\widehat{\boldsymbol{c}}$ from $\{0, 1\}^K$ to $[0, 1]^K$; similarly we relax the support of $\widehat{\boldsymbol{y}}$ from $\mathcal{Y} \subset \{0, 1\}^M$ to $[0, 1]^M$ (with the constraint that all entries of $\widehat{\boldsymbol{y}}$ sum up to 1). We use backpropagation to search for the optimal $\widehat{\boldsymbol{c}}$ and $\widehat{\boldsymbol{y}}$. After obtaining the optimal $\widehat{\boldsymbol{c}}$ and $\widehat{\boldsymbol{y}}$, we round them back to the binary vector space $\{0, 1\}^K$ and the one-hot vector space $\mathcal{Y}$ as the final prediction. More details are provided in Algorithm 1 of Appendix B. Comprehensive details about the hyperparameters used in this work can be found in Appendix B.1. Additionally, we present an ablation study that analyzes hyperparameter sensitivity in Table 5 of Appendix C.2.

### 3.3 Concept Intervention and Correction

Similar to most concept-based models, our ECBMs also supports test-time intervention. Specifically, after an ECBM predicts the concepts $\boldsymbol{c}$ and class label $\boldsymbol{y}$, practitioners can examine $\boldsymbol{c}$ and $\boldsymbol{y}$ to intervene on some of the concepts (e.g., correcting an incorrectly predicted concept). However, existing concept-based models do not capture the interaction between concepts; therefore correcting a concept does not help correct highly correlated concepts, leading to suboptimal concept and class accuracy. In contrast, our ECBMs are able to propagate the corrected concept(s) to other correlated concepts, thereby improving both concept and class accuracy. Proposition 3.1 below shows how our ECBMs automatically correct correlated concepts after test-time intervention and then leverage all corrected concepts to further improve final classification accuracy.

**Proposition 3.1** (**Joint Missing Concept and Class Probability**). *Given the ground-truth values of concepts $[c_k]_{k=1}^{K-s}$, the joint probability of the remaining concepts $[c_k]_{k=K-s+1}^{K}$ and the class label $\boldsymbol{y}$ can be computed as follows:*

$$p([c_k]_{k=K-s+1}^{K}, \boldsymbol{y}|\boldsymbol{x}, [c_k]_{k=1}^{K-s}) = \frac{e^{-E_{\boldsymbol{\theta}}^{joint}(\boldsymbol{x},\boldsymbol{c},\boldsymbol{y})}}{\sum_{m=1}^{M} \sum_{[c_k]_{K-s+1}^{K} \in \{0,1\}^s}(e^{-E_{\boldsymbol{\theta}}^{joint}(\boldsymbol{x},\boldsymbol{c},\boldsymbol{y}_m)})}, \tag{13}$$

*where $E_{\boldsymbol{\theta}}^{joint}(\boldsymbol{x}, \boldsymbol{c}, \boldsymbol{y})$ is the joint energy defined in Eqn. 12.*

### 3.4 Conditional Interpretations

ECBMs are capable of providing a range of conditional probabilities that effectively quantify the complex conditional dependencies between different concepts and class labels. These probabilities can be represented by energy levels. For example, Proposition 3.2 below computes $p(c_k|\boldsymbol{y})$ to interpret the importance of the concept $c_k$ to a specific class label $\boldsymbol{y}$ in an ECBM.

**Proposition 3.2** (**Marginal Class-Specific Concept Importance**). *Given the target class $\boldsymbol{y}$, the marginal concept importance (significance of each individual concept) can be expressed as:*

$$p(c_k|\boldsymbol{y}) \propto \sum_{\boldsymbol{c}_{\cdot k}} \frac{\sum_{\boldsymbol{x}} \left( \frac{e^{-E_{\boldsymbol{\theta}}^{global}(\boldsymbol{c},\boldsymbol{y})}}{\sum_{m=1}^{M} E_{\boldsymbol{\theta}}^{global}(\boldsymbol{c},\boldsymbol{y}_m)} \right) \cdot (e^{-\sum_{k'=1}^{K} E_{\boldsymbol{\theta}}^{concept}(\boldsymbol{x},c_{k'})}) \cdot p(\boldsymbol{x})}{\sum_{\boldsymbol{x}} e^{-E_{\boldsymbol{\theta}}^{class}(\boldsymbol{x},\boldsymbol{y})} \cdot p(\boldsymbol{x})}, \tag{14}$$

where $\boldsymbol{c}$ represents the full vector of concepts and can be broken down into $[c_k, \boldsymbol{c}_{-k}]$.

Proposition 3.2 above interprets the importance of each concept $c_k$ separately. In contrast, Proposition 3.3 below computes the joint distribution of all concepts $p(\boldsymbol{c}|\boldsymbol{y})$ to identify which combination of concepts $\boldsymbol{c}$ best represents a specific class $\boldsymbol{y}$.

**Proposition 3.3** (**Joint Class-Specific Concept Importance**). *Given the target class $\boldsymbol{y}$, the joint concept importance (significance of combined concepts) can be computed as:*

$$p(\boldsymbol{c}|\boldsymbol{y}) \propto \frac{\sum_{\boldsymbol{x}} \left( \frac{e^{-E_{\boldsymbol{\theta}}^{global}(\boldsymbol{c},\boldsymbol{y})}}{\sum_{m=1}^{M} E_{\boldsymbol{\theta}}^{global}(\boldsymbol{c},\boldsymbol{y}_m)} \right) \cdot (e^{-\sum_{k=1}^{K} E_{\boldsymbol{\theta}}^{concept}(\boldsymbol{x},c_k)}) \cdot p(\boldsymbol{x})}{\sum_{\boldsymbol{x}} e^{-E_{\boldsymbol{\theta}}^{class}(\boldsymbol{x},\boldsymbol{y})} \cdot p(\boldsymbol{x})}. \tag{15}$$

ECBMs can also provide interpretation on the probability of a correct concept prediction $c_k$, given the class label and another concept $c_{k'}$. This is computed as $p(c_k|c_{k'}, \boldsymbol{y})$ using Proposition 3.4 below. This demonstrates our ECBM's capability to reason about additional concepts when we have knowledge of specific labels and concepts.

**Proposition 3.4** (**Class-Specific Conditional Probability among Concepts**). *Given a concept label $c_{k'}$ and the class label $\boldsymbol{y}$, the probability of predicting another concept $c_k$ is:*

$$p(c_k|c_{k'}, \boldsymbol{y}) \propto \frac{\sum_{[c_j]_{j \neq k,k'}^{K} \in \{0,1\}^{K-2}} \sum_{\boldsymbol{x}} \left( \frac{e^{-E_{\boldsymbol{\theta}}^{global}(\boldsymbol{c},\boldsymbol{y})}}{\sum_{m=1}^{M} E_{\boldsymbol{\theta}}^{global}(\boldsymbol{c},\boldsymbol{y}_m)} \right) \cdot (e^{-\sum_{l=1}^{K} E_{\boldsymbol{\theta}}^{concept}(\boldsymbol{x},c_l)}) \cdot p(\boldsymbol{x})}{\sum_{[c_j]_{j \neq k}^{K} \in \{0,1\}^{K-1}} \sum_{\boldsymbol{x}} \left( \frac{e^{-E_{\boldsymbol{\theta}}^{global}(\boldsymbol{c},\boldsymbol{y})}}{\sum_{m=1}^{M} E_{\boldsymbol{\theta}}^{global}(\boldsymbol{c},\boldsymbol{y}_m)} \right) \cdot (e^{-\sum_{l=1}^{K} E_{\boldsymbol{\theta}}^{concept}(\boldsymbol{x},c_l)}) \cdot p(\boldsymbol{x})}.$$

Proposition 3.5 computes the conditional probability of one concept given another concept $p(c_k|c_{k'})$, which interprets the interaction (correlation) among concepts in an ECBM.

Table 1: Accuracy on Different Datasets. We report the mean and standard deviation from five runs with different random seeds. For ProbCBM (marked with "*"), we report the best results from the ProbCBM paper (Kim et al., 2023) for CUB and AWA2 datasets.

| Data Model | CUB | | | CelebA | | | AWA2 | | |
|---|---|---|---|---|---|---|---|---|---|
| Metric | Concept | Overall Concept | Class | Concept | Overall Concept | Class | Concept | Overall Concept | Class |
| CBM | $0.964 \pm 0.002$ | $0.364 \pm 0.070$ | $0.759 \pm 0.007$ | $0.837 \pm 0.009$ | $0.381 \pm 0.006$ | $0.246 \pm 0.005$ | $\mathbf{0.979} \pm 0.002$ | $0.803 \pm 0.023$ | $0.907 \pm 0.004$ |
| ProbCBM* | $0.946 \pm 0.001$ | $0.360 \pm 0.002$ | $0.718 \pm 0.005$ | $0.867 \pm 0.007$ | $0.473 \pm 0.001$ | $0.299 \pm 0.001$ | $0.959 \pm 0.000$ | $0.719 \pm 0.001$ | $0.880 \pm 0.001$ |
| PCBM | - | - | $0.635 \pm 0.002$ | - | - | $0.150 \pm 0.010$ | - | - | $0.862 \pm 0.003$ |
| CEM | $0.965 \pm 0.002$ | $0.396 \pm 0.052$ | $0.796 \pm 0.004$ | $0.867 \pm 0.001$ | $0.457 \pm 0.005$ | $0.330 \pm 0.003$ | $0.978 \pm 0.008$ | $0.796 \pm 0.011$ | $0.908 \pm 0.002$ |
| **ECBM** | $\mathbf{0.973} \pm 0.001$ | $\mathbf{0.713} \pm 0.009$ | $\mathbf{0.812} \pm 0.006$ | $\mathbf{0.876} \pm 0.000$ | $\mathbf{0.478} \pm 0.000$ | $\mathbf{0.343} \pm 0.000$ | $\mathbf{0.979} \pm 0.000$ | $\mathbf{0.854} \pm 0.000$ | $\mathbf{0.912} \pm 0.000$ |

**Proposition 3.5** (**Class-Agnostic Conditional Probability among Concepts**). *Given one concept $c_k$, the conditional probability of another concept $c_{k'}$ can be compuated as:*

$$p(c_k|c_{k'}) \propto \frac{\sum_{m=1}^{M} \sum_{[c_j]_{j\neq k,k'}^{K} \in \{0,1\}^{K-2}} \sum_{\boldsymbol{x}} \left( \frac{e^{-E_{\boldsymbol{\theta}}^{global}(\boldsymbol{c},\boldsymbol{y})}}{\sum_{m=1}^{M} E_{\boldsymbol{\theta}}^{global}(\boldsymbol{c},\boldsymbol{y}_m)} \right) \cdot (e^{-\sum_{l=1}^{K} E_{\boldsymbol{\theta}}^{concept}(\boldsymbol{x},c_l)}) \cdot p(\boldsymbol{x}) \cdot p(\boldsymbol{y}_m)}{\sum_{m=1}^{M} \sum_{[c_j]_{j\neq k}^{K} \in \{0,1\}^{K-1}} \sum_{\boldsymbol{x}} \left( \frac{e^{-E_{\boldsymbol{\theta}}^{global}(\boldsymbol{c},\boldsymbol{y})}}{\sum_{m=1}^{M} E_{\boldsymbol{\theta}}^{global}(\boldsymbol{c},\boldsymbol{y}_m)} \right) \cdot (e^{-\sum_{l=1}^{K} E_{\boldsymbol{\theta}}^{concept}(\boldsymbol{x},c_l)}) \cdot p(\boldsymbol{x}) \cdot p(\boldsymbol{y}_m)}.$$

Besides global interpretation above, ECBMs can also provide instance-level interpretation. For example, Proposition A.2 in Appendix A shows how ECBMs reason about the conditional probability of the class label $\boldsymbol{y}$ given the input $\boldsymbol{x}$ and a known concept $c_k$. **More ECBM conditional interpretations** and **all related proofs** are included in Appendix A.

# 4 EXPERIMENTS

In this section, we compare our ECBM with existing methods on real-world datasets.

## 4.1 EXPERIMENT SETUP

**Datasets.** We evaluate different methods on three real-world datasets:

- **Caltech-UCSD Birds-200-2011 (CUB)** (Wah et al., 2011) is a fine-grained bird classification dataset with 11,788 images, 200 classes and 312 annotated attributes. Following CBM (Koh et al., 2020), ProbCBM (Kim et al., 2023) and CEM (Zarlenga et al., 2022), we select 112 attributes as the concepts and use the same data splits.
- **Animals with Attributes 2 (AWA2)** (Xian et al., 2018) is a a zero-shot learning dataset containing 37,322 images and 50 animal classes. We use all 85 attributes as concepts.
- **Large-scale CelebFaces Attributes (CelebA)** (Liu et al., 2015) contains $200,000$ images, each annotated with 40 face attributes. Following the setting in CEM (Zarlenga et al., 2022), we use the 8 most balanced attributes as the target concepts and 256 classes for the classification task.

**Baselines and Implementation Details.** We compare our ECBM with state-of-the-art methods, i.e., concept bottleneck model (**CBM**) (Koh et al., 2020), concept embedding model (**CEM**) (Zarlenga et al., 2022), post-hoc concept bottleneck model (**PCBM**) (Yuksekgonul et al., 2022), and probabilistic concept bottleneck model (**ProbCBM**) (Kim et al., 2023). We use ResNet101 (He et al., 2016) as the feature extractor $F$ for all evaluated methods. We use the SGD optimizer during the training process. We use $\lambda_c = 0.3$ and $\lambda_g = 0.3$. For the propositions, we have implemented a hard version (yielding 0/1 output results) for computing probabilities. See Appendix B for more details.

**Evaluation Metrics.** With $\{\boldsymbol{x}^{(j)}, \boldsymbol{c}^{(j)}, \boldsymbol{y}^{(j)}\}_{j=1}^{N}$ as the dataset, we denote as $\{\widehat{\boldsymbol{c}}^{(j)}, \widehat{\boldsymbol{y}}^{(j)}\}_{j=1}^{N}$ the model prediction for concepts and class labels. $c_k^{(j)}$ and $\widehat{c}_k^{(j)}$ is the $k$-th dimension of $\boldsymbol{c}^{(j)}$ and $\widehat{\boldsymbol{c}}^{(j)}$, respectively. We use the following three metrics to evaluate different methods.

**Concept Accuracy** evaluates the model's predictions for each concept individually:

$$\mathcal{C}_{acc} = \sum_{j=1}^{N} \sum_{k=1}^{K} \mathbb{1}(c_k^{(j)} = \widehat{c}_k^{(j)}) / (KN), \tag{16}$$

where $\mathbb{1}(\cdot)$ is the indicator function.

**Overall Concept Accuracy** evaluates the model's ability to correctly predict *all* concepts for each input $\boldsymbol{x}^{(j)}$. Higher overall concept accuracy indicates the model's ability to mine the latent correlation between concepts for a more accurate interpretation for each concepts. It is defined as:

$$\mathcal{C}_{overall} = \sum_{j=1}^{N} \mathbb{1}(\boldsymbol{c}^{(j)} = \widehat{\boldsymbol{c}}^{(j)}) / N. \tag{17}$$

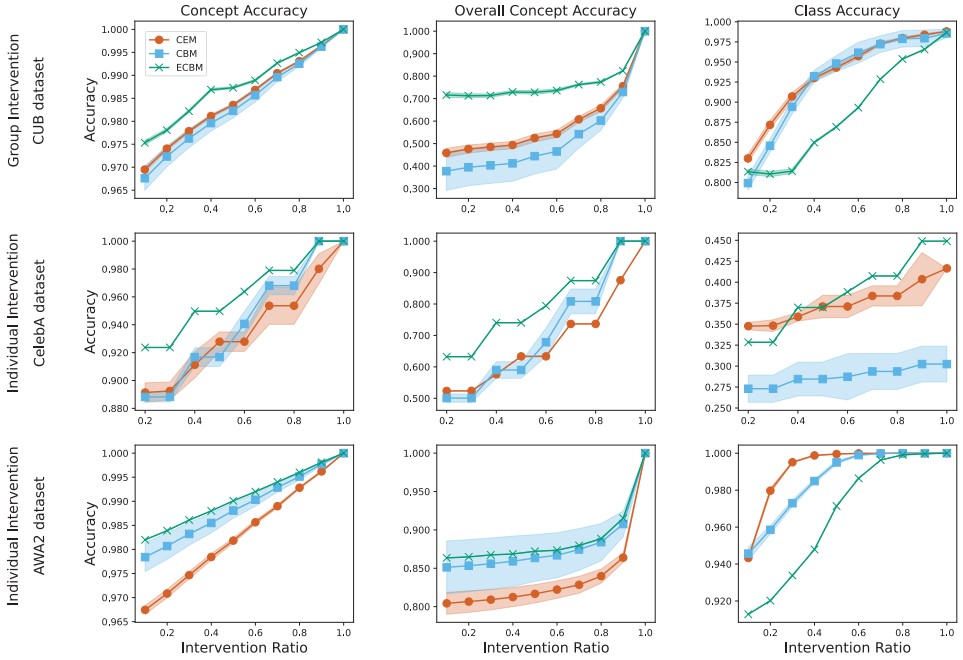

Figure 2: Performance with different ratios of intervened concepts on three datasets (with error bars). The intervention ratio denotes the proportion of provided correct concepts. We use CEM with RandInt. CelebA and AWA2 do not have grouped concepts; thus we adopt individual intervention.

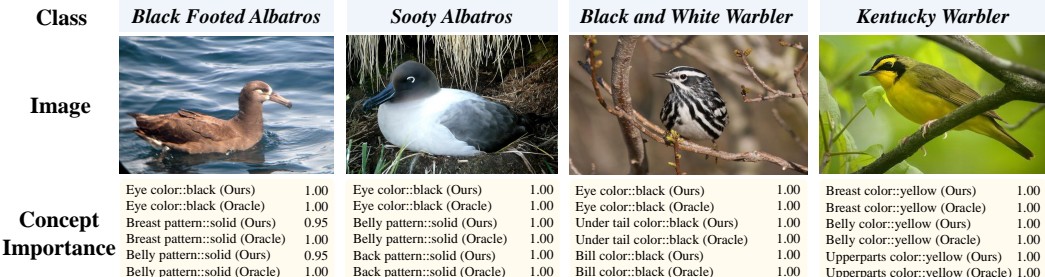

Figure 3: Marginal concept importance ($p(c_k = 1|\boldsymbol{y})$) for top 3 concepts of 4 different classes computed using Proposition 3.2. ECBM's estimation (Ours) is very close to the ground truth (Oracle).

**Class Accuracy** evaluates the model's prediction accuracy for the class label:

$$\mathcal{A}_{acc} = \sum_{j=1}^{N} \mathbb{1}(\boldsymbol{y}^{(j)} = \widehat{\boldsymbol{y}}^{(j)}) / N. \tag{18}$$

### 4.2 RESULTS

**Concept and Class Label Prediction.** Table 1 shows different types of accuracy of the evaluated methods. Concept accuracy across various methods is similar, with our ECBM slightly outperforming others. Interestingly, ECBM significantly outperforms other methods in terms of overall concept accuracy, especially in CUB (71.3% for ECBM versus 39.6% for the best baseline CEM); this shows that ECBM successfully captures the interaction (and correlation) among the concepts, thereby leveraging one correctly predicted concept to help correct other concepts' prediction. Such an advantage also helps improve ECBM's class accuracy upon other methods. We have conducted an ablation study for each component of our ECBM architecture (including a comparison with traditional black-box models) in Table 4 of Appendix C.2, verifying our design's effectiveness.

**Concept Intervention and Correction.** Problem Setting 2 in Sec. 3 and Proposition 3.1 introduce the scenario where a practitioner (e.g., a clinician) examine the predicted concepts (and class labels) and intervene on (correct) the concept prediction. An ideal model should leverage such intervention to automatically correct other concepts, thereby improving both interpretability and class prediction accuracy. Additional experiments (for the background shift dataset (Koh et al., 2020)) in the Appendix C.3 demonstrate the potential of our ECBM to enhance the robustness of CBMs. Fig. 2 shows three types of accuracy for different methods after intervening on (correcting) different proportions of the concepts, i.e., intervention ratios. In terms of both concept accuracy and overall concept

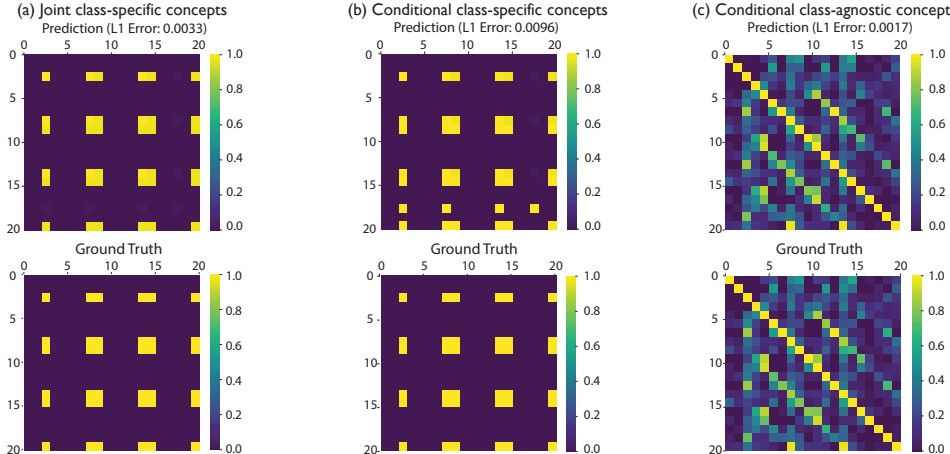

Figure 4: We selected the class "Black and White Warbler" in CUB for illustration. **(a)** Joint class-specific concept importance $p(c_{k'} = 1, c_k = 1|y)$ for ECBM's prediction and ground truth derived from Proposition 3.3. **(b)** Class-specific conditional probability among concepts $p(c_k = 1|c_{k'} = 1, y)$ for ECBM's prediction and ground truth derived from Proposition 3.4. **(c)** Class-agnostic conditional probability among concepts $p(c_k = 1|c_{k'} = 1)$ for ECBM's prediction and ground truth derived from Proposition 3.5.

accuracy, we can see that our ECBM outperforms the baselines across all intervention ratios. In terms of class accuracy, ECBM underperforms the vanilla CBM and the state-of-the-art CEM (with RandInt); this is because they have strict concept bottlenecks, and therefore even very few correct concepts can significantly improve class accuracy. Note that the primary focus of our ECBM is not class accuracy enhancement (detailed explanations and individual intervention on the CUB dataset (Fig. 12) can be found in Appendix C.5). We also provide further evidence demonstrating how our model can mitigate concept leakage in Fig. 11 of Appendix C.5.

**Conditional Interpretations.** Fig. 3 shows the marginal concept importance $(p(c_k|y))$ for top 3 concepts of 4 different classes, computed using Proposition 3.2. Our ECBM can provide interpretation on which concepts are the most important for predicting each class. For example, ECBM correctly identifies "eye color::black" and "bill color::black" as top concepts for "Black and White Warble"; for a similar class "Kentucky Warble", ECBM correctly identifies "breast color::yellow" and "belly color::yellow" as its top concepts. Quantitatively, ECBM's estimation (Ours) is very close to the ground truth (Oracle).

Fig. 4(a) and Fig. 4(b) show how ECBM interprets concept relations for a specific class. We show results for the first 20 concepts in CUB (see Table 3 in Appendix C for the concept list); we include full results (ECBM, CBM and CEM) on all 112 concepts in Appendix C. Specifically, Fig. 4(a) shows the joint class-specific concept importance, i.e., $p(c_{k'} = 1, c_k = 1|y)$ (with $y$ as "Black and White Warble"), computed using Proposition 3.3 versus the ground truth. For example, ECBM correctly estimates that for the class "Black and White Warble", concept "belly color" and "under tail color" have high joint probability; this is intuitive since different parts of a bird usually have the same color. Similarly, Fig. 4(b) shows class-specific conditional probability between different concepts, i.e., $p(c_k = 1|c_{k'} = 1, y)$ (with $y$ as "Black and White Warble"), computed using Proposition 3.4. Besides class-specific interpretation, Fig. 4(c) shows how ECBM interprets concept relations in general using conditional probability between concepts, i.e., $p(c_k|c_{k'})$, computed using Proposition 3.5. Quantitatively, the average L1 error (in the range $[0, 1]$) for Fig. 4(a-c) is 0.0033, 0.0096, and 0.0017, respectively, demonstrating ECBM's accurate conditional interpretation.

## 5 CONCLUSION AND LIMITATIONS

In this paper, we go beyond typical concept-based prediction to identify the problems of concept correction and conditional interpretation as valuable tools to provide concept-based interpretations. We propose ECBM, the first general method to unify concept-based prediction, concept correction, and conditional interpretation as conditional probabilities under a joint energy formulation. Future work may include extending ECBM to handle uncertainty quantification using Bayesian neural networks (Wang & Wang, 2023), enable unsupervised learning of concepts (Ma et al., 2023) via graphical models within the hierarchical Bayesian deep learning framework (Wang & Yeung, 2016; 2020), and enable cross-domain interpretation (Wang et al., 2020; Xu et al., 2022; Liu et al., 2023b).

ACKNOWLEDGMENT

The authors thank the reviewers/ACs for the constructive comments to improve the paper. The authors are also grateful to Min Shi and Yueying Hu for their comments to improve this paper. This work is supported in part by the National Natural Science Foundation of China under Grant 62306254 and in part by the Hong Kong Innovation and Technology Fund under Grant ITS/030/21. Xinyue Xu is supported by the Hong Kong PhD Fellowship Scheme (HKPFS) from Hong Kong Research Grants Council (RGC).

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

## A   ADDITIONAL CONDITIONAL INTERPRETATIONS AND PROOFS

Given the input $\boldsymbol{x}$ and label $\boldsymbol{y}$, we propose to use the Boltzmann distribution to define the conditional likelihood of label $\boldsymbol{y}$ given $\boldsymbol{x}$:

$$p_{\boldsymbol{\theta}}(\boldsymbol{y}|\boldsymbol{x}) = \frac{\exp\big(-E_{\boldsymbol{\theta}}^{class}(\boldsymbol{x},\boldsymbol{y})\big)}{\sum_{m=1}^{M}\exp\big(-E_{\boldsymbol{\theta}}^{class}(\boldsymbol{x},\boldsymbol{y}_m)\big)}, \tag{19}$$

where $\boldsymbol{y}_m \in \mathcal{Y}$ a one-hot vector with the $m$-th dimension set to 1.

We use the negative log-likelihood as the loss function for an input-class pair $(\boldsymbol{x}, \boldsymbol{y})$, and the expanded form is:

$$
\begin{aligned}
\mathcal{L}_{class}(\boldsymbol{x}, \boldsymbol{y}) &= -\log p_{\boldsymbol{\theta}}(\boldsymbol{y}|\boldsymbol{x}) \\
&= -\log \frac{\exp\left(-E_{\boldsymbol{\theta}}^{class}(\boldsymbol{x}, \boldsymbol{y})\right)}{\sum_{m=1}^{M} \exp\left(-E_{\boldsymbol{\theta}}^{class}(\boldsymbol{x}, \boldsymbol{y}_m)\right)} \\
&= -\log\left(\exp\left(-E_{\boldsymbol{\theta}}^{class}(\boldsymbol{x}, \boldsymbol{y})\right)\right) + \log\left(\sum_{m=1}^{M} \exp\left(-E_{\boldsymbol{\theta}}^{class}(\boldsymbol{x}, \boldsymbol{y}_m)\right)\right) \\
&= E_{\boldsymbol{\theta}}^{class}(\boldsymbol{x}, \boldsymbol{y}) + \log\left(\sum_{m=1}^{M} e^{-E_{\boldsymbol{\theta}}^{class}(\boldsymbol{x}, \boldsymbol{y}_m)}\right).
\end{aligned}
\tag{20}
$$

Thus, we can have:

$$
p(\boldsymbol{y}|\boldsymbol{x}) \propto e^{-E_{\boldsymbol{\theta}}^{class}(\boldsymbol{x}, \boldsymbol{y})},
\tag{21}
$$

which connect our energy function $E_{\boldsymbol{\theta}}^{class}(\boldsymbol{x}, \boldsymbol{y})$ to the conditional probability $p(\boldsymbol{y}|\boldsymbol{x})$.

Similarly, we denote the local concept energy of the energy network parameterized by $\boldsymbol{\theta}$ between input $\boldsymbol{x}$ and the $k$-th dimension of concept $c_k$ as $E_{\boldsymbol{\theta}}^{concept}(\boldsymbol{x}, c_k)$. We can obtain:

$$
p(c_k|\boldsymbol{x}) \propto e^{-E_{\boldsymbol{\theta}}^{concept}(\boldsymbol{x}, c_k)}.
\tag{22}
$$

We denote the global concept-class energy of the energy network parameterized by $\boldsymbol{\theta}$ between input $\boldsymbol{c}$ and the label $\boldsymbol{y}$ as $E_{\boldsymbol{\theta}}^{global}(\boldsymbol{c}, \boldsymbol{y})$. Similarly, we then have:

$$
p(\boldsymbol{y}|\boldsymbol{c}) = \frac{e^{-E_{\boldsymbol{\theta}}^{global}(\boldsymbol{c}, \boldsymbol{y})}}{\sum_{m=1}^{M} e^{-E_{\boldsymbol{\theta}}^{global}(\boldsymbol{c}, \boldsymbol{y}_m)}}.
\tag{23}
$$

Similarly, with the joint energy in Eqn. 12, we have

$$
p(\boldsymbol{x}, \boldsymbol{c}, \boldsymbol{y}) = \frac{e^{-E_{\boldsymbol{\theta}}^{joint}(\boldsymbol{x}, \boldsymbol{c}, \boldsymbol{y})}}{\sum_{m=1}^{M} e^{-E_{\boldsymbol{\theta}}^{joint}(\boldsymbol{x}, \boldsymbol{c}, \boldsymbol{y}_m)}}.
\tag{24}
$$

**Proposition 3.1** (**Joint Missing Concept and Class Probability**). *Given the ground-truth values of concepts $[c_k]_{k=1}^{K-s}$, the joint probability of the remaining concepts $[c_k]_{k=K-s+1}^{K}$ and the class label $\boldsymbol{y}$ can be computed as follows:*

$$
p([c_k]_{k=K-s+1}^{K}, \boldsymbol{y}|\boldsymbol{x}, [c_k]_{k=1}^{K-s}) = \frac{e^{-E_{\boldsymbol{\theta}}^{joint}(\boldsymbol{x}, \boldsymbol{c}, \boldsymbol{y})}}{\sum_{m=1}^{M} \sum_{[c_k]_{K-s+1}^{K} \in \{0,1\}^s} (e^{-E_{\boldsymbol{\theta}}^{joint}(\boldsymbol{x}, \boldsymbol{c}, \boldsymbol{y}_m)})},
\tag{13}
$$

*where $E_{\boldsymbol{\theta}}^{joint}(\boldsymbol{x}, \boldsymbol{c}, \boldsymbol{y})$ is the joint energy defined in Eqn. 12.*

*Proof.* By definition of joint energy, we have that

$$
p([c_k]_{k=K-s+1}^{K}, \boldsymbol{y}|\boldsymbol{x}, [c_k]_{k=1}^{K-s}) \propto e^{-E_{\boldsymbol{\theta}}^{joint}(\boldsymbol{x}, \boldsymbol{c}, \boldsymbol{y})}.
\tag{25}
$$

Therefore by Bayes rule, we then have

$$
p([c_k]_{k=K-s+1}^{K}, \boldsymbol{y}|\boldsymbol{x}, [c_k]_{k=1}^{K-s}) = \frac{e^{-E_{\boldsymbol{\theta}}^{joint}(\boldsymbol{x}, \boldsymbol{c}, \boldsymbol{y})}}{\sum_{m=1}^{M} \sum_{[c_k]_{K-s+1}^{K} \in \{0,1\}^s} (e^{-E_{\boldsymbol{\theta}}^{joint}(\boldsymbol{x}, \boldsymbol{c}, \boldsymbol{y}_m)})},
\tag{26}
$$

concluding the proof. $\square$

We initially establish the Joint Class-Specific Concept Importance (Proposition 3.3), which we subsequently employ to marginalize $c_k$ and demonstrate the Marginal Class-Specific Concept Importance (Proposition 3.2).

**Proposition 3.3** (**Joint Class-Specific Concept Importance**). *Given the target class $\boldsymbol{y}$, the joint concept importance (significance of combined concepts) can be computed as:*

$$
p(\boldsymbol{c}|\boldsymbol{y}) \propto \frac{\sum_{\boldsymbol{x}}\left(\frac{e^{-E_{\boldsymbol{\theta}}^{global}(\boldsymbol{c}, \boldsymbol{y})}}{\sum_{m=1}^{M} E_{\boldsymbol{\theta}}^{global}(\boldsymbol{c}, \boldsymbol{y}_m)}\right) \cdot (e^{-\sum_{k=1}^{K} E_{\boldsymbol{\theta}}^{concept}(\boldsymbol{x}, c_k)}) \cdot p(\boldsymbol{x})}{\sum_{\boldsymbol{x}} e^{-E_{\boldsymbol{\theta}}^{class}(\boldsymbol{x}, \boldsymbol{y}) \cdot p(\boldsymbol{x})}}.
\tag{15}
$$

*Proof.* Given Eqn. 21, Eqn. 22 and Eqn. 23, we have

$$
\begin{aligned}
p(\boldsymbol{c}|\boldsymbol{y}) &= \frac{p(\boldsymbol{y}|\boldsymbol{c}) \cdot p(\boldsymbol{c})}{p(\boldsymbol{y})} \\
&= \frac{\sum_{\boldsymbol{x}} p(\boldsymbol{y}|\boldsymbol{c}) \cdot p(\boldsymbol{c}|\boldsymbol{x}) \cdot p(\boldsymbol{x})}{p(\boldsymbol{y})} \\
&= \frac{\sum_{\boldsymbol{x}} p(\boldsymbol{y}|\boldsymbol{c}) \cdot (\prod_{k=1}^{K} p(c_k|\boldsymbol{x})) \cdot p(\boldsymbol{x})}{p(\boldsymbol{y})} \\
&= \frac{\sum_{\boldsymbol{x}} \left( \frac{e^{-E_{\boldsymbol{\theta}}^{global}(\boldsymbol{c},\boldsymbol{y})}}{\sum_{m=1}^{M} E_{\boldsymbol{\theta}}^{global}(\boldsymbol{c},\boldsymbol{y}_m)} \right) \cdot (\prod_{k=1}^{K} p(c_k|\boldsymbol{x})) \cdot p(\boldsymbol{x})}{\sum_{\boldsymbol{x}} p(\boldsymbol{y}|\boldsymbol{x}) \cdot p(\boldsymbol{x})} \\
&\propto \frac{\sum_{\boldsymbol{x}} \left( \frac{e^{-E_{\boldsymbol{\theta}}^{global}(\boldsymbol{c},\boldsymbol{y})}}{\sum_{m=1}^{M} E_{\boldsymbol{\theta}}^{global}(\boldsymbol{c},\boldsymbol{y}_m)} \right) \cdot (e^{-\sum_{k=1}^{K} E_{\boldsymbol{\theta}}^{concept}(\boldsymbol{x},c_k)}) \cdot p(\boldsymbol{x})}{\sum_{\boldsymbol{x}} e^{-E_{\boldsymbol{\theta}}^{class}(\boldsymbol{x},\boldsymbol{y})} \cdot p(\boldsymbol{x})},
\end{aligned}
\tag{27}
$$

where

$$
\begin{aligned}
p(\boldsymbol{y}) &= \sum_{x} p(\boldsymbol{x},\boldsymbol{y}) \\
&= \sum_{\boldsymbol{x}} p(\boldsymbol{y}|\boldsymbol{x}) \cdot p(\boldsymbol{x}) \\
&= \sum_{\boldsymbol{x}} e^{-E_{\boldsymbol{\theta}}^{class}(\boldsymbol{x},\boldsymbol{y})} \cdot p(\boldsymbol{x}), \\
or \\
&= \sum_{\boldsymbol{c}} p(\boldsymbol{y}|\boldsymbol{c}) \cdot p(\boldsymbol{c}) \\
&= \sum_{\boldsymbol{c}} (\frac{e^{-E_{\boldsymbol{\theta}}^{global}(\boldsymbol{c},\boldsymbol{y})}}{\sum_{m=1}^{M} E_{\boldsymbol{\theta}}^{global}(\boldsymbol{c},\boldsymbol{y}_m)}) \cdot p(\boldsymbol{c}),
\end{aligned}
\tag{28}
$$

concluding the proof. $\square$

**Proposition 3.2 (Marginal Class-Specific Concept Importance).** *Given the target class $\boldsymbol{y}$, the marginal concept importance (significance of each individual concept) can be expressed as:*

$$
p(c_k|\boldsymbol{y}) \propto \sum_{\boldsymbol{c}_{\text{-}k}} \frac{\sum_{\boldsymbol{x}} \left( \frac{e^{-E_{\boldsymbol{\theta}}^{global}(\boldsymbol{c},\boldsymbol{y})}}{\sum_{m=1}^{M} E_{\boldsymbol{\theta}}^{global}(\boldsymbol{c},\boldsymbol{y}_m)} \right) \cdot (e^{-\sum_{k'=1}^{K} E_{\boldsymbol{\theta}}^{concept}(\boldsymbol{x},c_{k'})}) \cdot p(\boldsymbol{x})}{\sum_{\boldsymbol{x}} e^{-E_{\boldsymbol{\theta}}^{class}(\boldsymbol{x},\boldsymbol{y})} \cdot p(\boldsymbol{x})},
\tag{14}
$$

*Proof.* Given Eqn. 21, Eqn. 22, Eqn. 23 and Proposition 3.3, marginal class-specific concept importance is marginalize $c_k \in \boldsymbol{c}$ of Proposition 3.3, we have

$$
\begin{aligned}
p(c_k|\boldsymbol{y}) &= \sum_{\boldsymbol{c}_{\text{-}k}} p(c_k, \boldsymbol{c}_{\text{-}k}|\boldsymbol{y}) \\
&\propto \sum_{\boldsymbol{c}_{\text{-}k}} \frac{\sum_{\boldsymbol{x}} \left( \frac{e^{-E_{\boldsymbol{\theta}}^{global}(\boldsymbol{c},\boldsymbol{y})}}{\sum_{m=1}^{M} E_{\boldsymbol{\theta}}^{global}(\boldsymbol{c},\boldsymbol{y}_m)} \right) \cdot (e^{-\sum_{k'=1}^{K} E_{\boldsymbol{\theta}}^{concept}(\boldsymbol{x},c_{k'})}) \cdot p(\boldsymbol{x})}{\sum_{\boldsymbol{x}} e^{-E_{\boldsymbol{\theta}}^{class}(\boldsymbol{x},\boldsymbol{y})} \cdot p(\boldsymbol{x})},
\end{aligned}
\tag{29}
$$

concluding the proof. $\square$

**Proposition 3.4 (Class-Specific Conditional Probability among Concepts).** *Given a concept label $c_{k'}$ and the class label $\boldsymbol{y}$, the probability of predicting another concept $c_k$ is:*

$$
p(c_k|c_{k'}, \boldsymbol{y}) \propto \frac{\sum_{[c_j]_{j\neq k,k'}^{K} \in \{0,1\}^{K-2}} \sum_{\boldsymbol{x}} \left( \frac{e^{-E_{\boldsymbol{\theta}}^{global}(\boldsymbol{c},\boldsymbol{y})}}{\sum_{m=1}^{M} E_{\boldsymbol{\theta}}^{global}(\boldsymbol{c},\boldsymbol{y}_m)} \right) \cdot (e^{-\sum_{l=1}^{K} E_{\boldsymbol{\theta}}^{concept}(\boldsymbol{x},c_l)}) \cdot p(\boldsymbol{x})}{\sum_{[c_j]_{j\neq k}^{K} \in \{0,1\}^{K-1}} \sum_{\boldsymbol{x}} \left( \frac{e^{-E_{\boldsymbol{\theta}}^{global}(\boldsymbol{c},\boldsymbol{y})}}{\sum_{m=1}^{M} E_{\boldsymbol{\theta}}^{global}(\boldsymbol{c},\boldsymbol{y}_m)} \right) \cdot (e^{-\sum_{l=1}^{K} E_{\boldsymbol{\theta}}^{concept}(\boldsymbol{x},c_l)}) \cdot p(\boldsymbol{x})}.
$$

*Proof.* Given Eqn. 22, Eqn. 23, and Proposition 3.3, we have

$$
\begin{aligned}
p(c_k|c_{k'}, \boldsymbol{y}) &= \frac{p(c_k, c_{k'}|\boldsymbol{y})}{p(c_{k'}|\boldsymbol{y})} \\
&= \frac{\sum_{[c_j]_{j\neq k,k'}^K \in \{0,1\}^{K-2}} p(\boldsymbol{c}|\boldsymbol{y})}{\sum_{[c_j]_{j\neq k}^K \in \{0,1\}^{K-1}} p(\boldsymbol{c}|\boldsymbol{y})} \\
&= \frac{\sum_{[c_j]_{j\neq k,k'}^K \in \{0,1\}^{K-2}} p(c_k, c_{k'}, [c_j]_{j\neq k,k'}^K | \boldsymbol{y})}{\sum_{[c_j]_{j\neq k}^K \in \{0,1\}^{K-1}} p(c_k, \boldsymbol{c}_{-k}|\boldsymbol{y})} \\
&\propto \frac{\sum_{[c_j]_{j\neq k,k'}^K \in \{0,1\}^{K-2}} \frac{\sum_{\boldsymbol{x}}\left(\frac{e^{-E_{\boldsymbol{\theta}}^{global}(\boldsymbol{c},\boldsymbol{y})}}{\sum_{m=1}^M E_{\boldsymbol{\theta}}^{global}(\boldsymbol{c},\boldsymbol{y}_m)}\right)\cdot(e^{-\sum_{l=1}^K E_{\boldsymbol{\theta}}^{concept}(\boldsymbol{x},c_l)})\cdot p(\boldsymbol{x})}{\sum_{\boldsymbol{x}} e^{-E_{\boldsymbol{\theta}}^{class}(\boldsymbol{x},\boldsymbol{y})}\cdot p(\boldsymbol{x})}}{\sum_{[c_j]_{j\neq k}^K \in \{0,1\}^{K-1}} \frac{\sum_{\boldsymbol{x}}\left(\frac{e^{-E_{\boldsymbol{\theta}}^{global}(\boldsymbol{c},\boldsymbol{y})}}{\sum_{m=1}^M E_{\boldsymbol{\theta}}^{global}(\boldsymbol{c},\boldsymbol{y}_m)}\right)\cdot(e^{-\sum_{l=1}^K E_{\boldsymbol{\theta}}^{concept}(\boldsymbol{x},c_l)})\cdot p(\boldsymbol{x})}{\sum_{\boldsymbol{x}} e^{-E_{\boldsymbol{\theta}}^{class}(\boldsymbol{x},\boldsymbol{y})}\cdot p(\boldsymbol{x})}} \\
&\propto \frac{\sum_{[c_j]_{j\neq k,k'}^K \in \{0,1\}^{K-2}} \sum_{\boldsymbol{x}}\left(\frac{e^{-E_{\boldsymbol{\theta}}^{global}(\boldsymbol{c},\boldsymbol{y})}}{\sum_{m=1}^M E_{\boldsymbol{\theta}}^{global}(\boldsymbol{c},\boldsymbol{y}_m)}\right)\cdot(e^{-\sum_{l=1}^K E_{\boldsymbol{\theta}}^{concept}(\boldsymbol{x},c_l)})\cdot p(\boldsymbol{x})}{\sum_{[c_j]_{j\neq k}^K \in \{0,1\}^{K-1}} \sum_{\boldsymbol{x}}\left(\frac{e^{-E_{\boldsymbol{\theta}}^{global}(\boldsymbol{c},\boldsymbol{y})}}{\sum_{m=1}^M E_{\boldsymbol{\theta}}^{global}(\boldsymbol{c},\boldsymbol{y}_m)}\right)\cdot(e^{-\sum_{l=1}^K E_{\boldsymbol{\theta}}^{concept}(\boldsymbol{x},c_l)})\cdot p(\boldsymbol{x})},
\end{aligned}
\tag{30}
$$

concluding the proof. $\qquad\square$

**Proposition 3.5** (**Class-Agnostic Conditional Probability among Concepts**). *Given one concept $c_k$, the conditional probability of another concept $c_{k'}$ can be computed as:*

$$
p(c_k|c_{k'}) \propto \frac{\sum_{m=1}^M \sum_{[c_j]_{j\neq k,k'}^K \in \{0,1\}^{K-2}} \sum_{\boldsymbol{x}}\left(\frac{e^{-E_{\boldsymbol{\theta}}^{global}(\boldsymbol{c},\boldsymbol{y})}}{\sum_{m=1}^M E_{\boldsymbol{\theta}}^{global}(\boldsymbol{c},\boldsymbol{y}_m)}\right)\cdot(e^{-\sum_{l=1}^K E_{\boldsymbol{\theta}}^{concept}(\boldsymbol{x},c_l)})\cdot p(\boldsymbol{x})\cdot p(\boldsymbol{y}_m)}{\sum_{m=1}^M \sum_{[c_j]_{j\neq k}^K \in \{0,1\}^{K-1}} \sum_{\boldsymbol{x}}\left(\frac{e^{-E_{\boldsymbol{\theta}}^{global}(\boldsymbol{c},\boldsymbol{y})}}{\sum_{m=1}^M E_{\boldsymbol{\theta}}^{global}(\boldsymbol{c},\boldsymbol{y}_m)}\right)\cdot(e^{-\sum_{l=1}^K E_{\boldsymbol{\theta}}^{concept}(\boldsymbol{x},c_l)})\cdot p(\boldsymbol{x})\cdot p(\boldsymbol{y}_m)}.
$$

*Proof.* Given Proposition 3.3 and Propostion 3.4, we have

$$
\begin{aligned}
p(c_k|c_{k'}) &= \frac{p(c_k, c_{k'})}{p(c_{k'})} \\
&= \frac{\sum_{m=1}^M p(c_k, c_{k'}|\boldsymbol{y}_m)\cdot p(\boldsymbol{y}_m)}{\sum_{m=1}^M p(c_{k'}|\boldsymbol{y}_m)\cdot p(\boldsymbol{y}_m)} \\
&= \frac{\sum_{m=1}^M \sum_{[c_j]_{j\neq k,k'}^K \in \{0,1\}^{K-2}} p(c_k, c_{k'}, [c_j]_{j\neq k,k'}^K|\boldsymbol{y}_m)\cdot p(\boldsymbol{y}_m)}{\sum_{m=1}^M \sum_{[c_j]_{j\neq k}^K \in \{0,1\}^{K-1}} p(c_k, \boldsymbol{c}_{-k}|\boldsymbol{y}_m)\cdot p(\boldsymbol{y}_m)} \\
&\propto \frac{\sum_{m=1}^M \sum_{[c_j]_{j\neq k,k'}^K \in \{0,1\}^{K-2}} \sum_{\boldsymbol{x}}\left(\frac{e^{-E_{\boldsymbol{\theta}}^{global}(\boldsymbol{c},\boldsymbol{y})}}{\sum_{m=1}^M E_{\boldsymbol{\theta}}^{global}(\boldsymbol{c},\boldsymbol{y}_m)}\right)\cdot(e^{-\sum_{l=1}^K E_{\boldsymbol{\theta}}^{concept}(\boldsymbol{x},c_l)})\cdot p(\boldsymbol{x})\cdot p(\boldsymbol{y}_m)}{\sum_{m=1}^M \sum_{[c_j]_{j\neq k}^K \in \{0,1\}^{K-1}} \sum_{\boldsymbol{x}}\left(\frac{e^{-E_{\boldsymbol{\theta}}^{global}(\boldsymbol{c},\boldsymbol{y})}}{\sum_{m=1}^M E_{\boldsymbol{\theta}}^{global}(\boldsymbol{c},\boldsymbol{y}_m)}\right)\cdot(e^{-\sum_{l=1}^K E_{\boldsymbol{\theta}}^{concept}(\boldsymbol{x},c_l)})\cdot p(\boldsymbol{x})\cdot p(\boldsymbol{y}_m)},
\end{aligned}
\tag{31}
$$

concluding the proof. $\qquad\square$

**Proposition A.1** (**Missing Concept Probability**). *Given the ground-truth values of concepts $[c_k]_{k=1}^{K-s}$, the joint probability of the remaining concepts $c_{k'}$ can be computed as follows:*

$$
p([c_k]_{k=K-s+1}^K|\boldsymbol{x}, [c_k]_{k=1}^{K-s}) = \frac{\sum_{m=1}^M e^{-E_{\boldsymbol{\theta}}^{joint}(\boldsymbol{x},\boldsymbol{c},\boldsymbol{y}_m)}}{\sum_{m=1}^M \sum_{[c_k]_{K-s+1}^K \in \{0,1\}^s}(e^{-E_{\boldsymbol{\theta}}^{joint}(\boldsymbol{x},\boldsymbol{c},\boldsymbol{y}_m)})},
\tag{32}
$$

*where $E_{\boldsymbol{\theta}}^{joint}(\boldsymbol{x}, \boldsymbol{c}, \boldsymbol{y})$ is the joint energy defined in Eqn. 12.*

*Proof.* This follows directly after marginalizing $\boldsymbol{y}$ out from $p([c_k]_{k=K-s+1}^K, \boldsymbol{y}|\boldsymbol{x}, [c_k]_{k=1}^{K-s})$ in Proposition 3.1. $\qquad\square$

**Proposition A.2** (**Conditional Class Probability Given a Known Concept**). *Given the input $\boldsymbol{x}$ and a concept $c_k$, the conditional probability of label $\boldsymbol{y}$ is:*

$$p(\boldsymbol{y}|\boldsymbol{x}, c_k) \propto \frac{\sum_{\boldsymbol{c}_{\cdot k}} \frac{e^{-E_{\boldsymbol{\theta}}^{joint}(\boldsymbol{x}, \boldsymbol{c}, \boldsymbol{y})}}{\sum_{m=1}^{M} e^{-E_{\boldsymbol{\theta}}^{joint}(\boldsymbol{x}, \boldsymbol{c}, \boldsymbol{y}_m)}}}{e^{-E_{\boldsymbol{\theta}}^{concept}(\boldsymbol{x}, c_k)}}. \tag{33}$$

*Proof.* Given Eqn. 22 and Eqn. 24, marginalize $c_k \in \boldsymbol{c}$ of Eqn. 24, we have

$$
\begin{aligned}
p(\boldsymbol{y}|\boldsymbol{x}, c_k) &= \frac{p(\boldsymbol{y}, \boldsymbol{x}, c_k)}{p(\boldsymbol{x}, c_k)} \\
&= \frac{\sum_{\boldsymbol{c}_{\cdot k}} p(\boldsymbol{y}, \boldsymbol{x}, \boldsymbol{c})}{p(\boldsymbol{x}, c_k)} \\
&= \frac{\sum_{\boldsymbol{c}_{\cdot k}} p(\boldsymbol{y}, \boldsymbol{x}, c_k, \boldsymbol{c}_{\cdot k})}{p(\boldsymbol{x}, c_k)} \\
&= \frac{\sum_{\boldsymbol{c}_{\cdot k}} \frac{e^{-E_{\boldsymbol{\theta}}^{joint}(\boldsymbol{x}, \boldsymbol{c}, \boldsymbol{y})}}{\sum_{m=1}^{M} e^{-E_{\boldsymbol{\theta}}^{joint}(\boldsymbol{x}, \boldsymbol{c}, \boldsymbol{y}_m)}}}{p(c_k|\boldsymbol{x}) \cdot p(\boldsymbol{x})} \\
&\propto \frac{\sum_{\boldsymbol{c}_{\cdot k}} \frac{e^{-E_{\boldsymbol{\theta}}^{joint}(\boldsymbol{x}, \boldsymbol{c}, \boldsymbol{y})}}{\sum_{m=1}^{M} e^{-E_{\boldsymbol{\theta}}^{joint}(\boldsymbol{x}, \boldsymbol{c}, \boldsymbol{y}_m)}}}{e^{-E_{\boldsymbol{\theta}}^{concept}(\boldsymbol{x}, c_k)}},
\end{aligned}
\tag{34}
$$

concluding the proof. □

# B   IMPLEMENTATION DETAILS

## B.1   HYPERPARAMETERS

For a fair comparison, we use ResNet101 (He et al., 2016) as the backbone and $299 \times 299$ as the input size for all evaluated methods, except CelebA with $64 \times 64$. We use the SGD optimizer to train the model. We use $\lambda_c = 0.3$, $\lambda_g = 0.3$, batch size 64, a learning rate of $1 \times 10^{-2}$, and at most 300 epochs. In the gradient inference process, we use the Adam optimizer, $\lambda_c = 0.49$, $\lambda_g = 0.004$ (this is equivalent to applying a ratio of 1:1:0.01 for the class, concept and global energy terms, respectively) , batch size 64, and a learning rate of $1 \times 10^{-1}$. We run all experiments on an NVIDIA RTX3090 GPU. In order to enhance the robustness of the $\boldsymbol{c}$-$\boldsymbol{y}$ energy head, we deployed perturbation augmentation when training ECBMs. Specifically, we perturbed $20\%$ of noisy pairs at probability $p = 0.2$ at the input of $\boldsymbol{c}$-$\boldsymbol{y}$ energy head during the training phase. These are not incorporated during the phases of inference and intervention.

Regarding hyperparameter $\lambda_c = 0.49$, this is because in our implementation, we introduce $\lambda_l$ to Eq (12) in the paper, resulting in: $\lambda_l E_{\theta}^{class}(\boldsymbol{x}, \boldsymbol{y}) + \lambda_c E_{\theta}^{concept}(\boldsymbol{x}, \boldsymbol{c}) + \lambda_g E_{\theta}^{global}(\boldsymbol{c}, \boldsymbol{y})$. We then set $\lambda_l = 1$, $\lambda_c = 1$ and $\lambda_g = 0.01$. This is therefore equivalent to setting $\lambda_c = 1/(1 + 1 + 0.01) = 0.49$ for the original Eq (12) without $\lambda_l$: $E_{\theta}^{class}(\boldsymbol{x}, \boldsymbol{y}) + \lambda_c E_{\theta}^{concept}(\boldsymbol{x}, \boldsymbol{c}) + \lambda_g E_{\theta}^{global}(\boldsymbol{c}, \boldsymbol{y})$.

---

**Algorithm 1** Gradient-Based Inference Algorithm

---

1: **Input:** Image $\boldsymbol{x}$, positive and negative concept embedding $[\boldsymbol{v}_k^{(+)}]_{k=1}^K$ and $[\boldsymbol{v}_k^{(-)}]_{k=1}^K$, label embedding $[\boldsymbol{u}_m]_{m=1}^M$, weight parameters $\lambda_c, \lambda_g$, learning rate $\eta$, concept and class number $K, M$.

2: **Output:** Concept and label probability $\widehat{\boldsymbol{c}}, \widehat{\boldsymbol{y}}$.
3: Initialize un-normalized concept probability $\widetilde{\boldsymbol{c}}$.
4: Initialize un-normalized class probability $\widetilde{\boldsymbol{y}}$.
5: **while** not converge **do**
6:     $\widehat{\boldsymbol{c}} \leftarrow Sigmoid(\widetilde{\boldsymbol{c}})$
7:     $\widehat{\boldsymbol{y}} \leftarrow Softmax(\widetilde{\boldsymbol{y}})$
8:     **for** $k \leftarrow 1$ **To** $K$ **do**
9:         $\boldsymbol{v}_k' \leftarrow \widehat{\boldsymbol{c}}_k \times \boldsymbol{v}_k^{(+)} + (1 - \widehat{\boldsymbol{c}}_k) \times \boldsymbol{v}_k^{(-)}$
10:     **end for**
11:     **for** $m \leftarrow 1$ **To** $M$ **do**
12:         $\boldsymbol{u}_m' \leftarrow \widehat{\boldsymbol{y}}_m \times \boldsymbol{u}_m$
13:     **end for**
14:     Calculate $\mathcal{L}_{class}(\boldsymbol{x}, \widehat{\boldsymbol{y}})$ based on Eqn. 5.
15:     Calculate $\mathcal{L}_{concept}(\boldsymbol{x}, \widehat{\boldsymbol{c}})$ based on Eqn. 8.
16:     Calculate $\mathcal{L}_{global}(\widehat{\boldsymbol{c}}, \widehat{\boldsymbol{y}})$ based on Eqn. 10.
17:     $\mathcal{L}_{total} = \mathcal{L}_{class} + \lambda_c \mathcal{L}_{concept} + \lambda_g \mathcal{L}_{global}$
18:     $\widetilde{\boldsymbol{c}} \leftarrow \widetilde{\boldsymbol{c}} - \eta \nabla \mathcal{L}_{total}$
19:     $\widetilde{\boldsymbol{y}} \leftarrow \widetilde{\boldsymbol{y}} - \eta \nabla \mathcal{L}_{total}$
20: **end while**
21: $\widehat{\boldsymbol{c}} \leftarrow Sigmoid(\widetilde{\boldsymbol{c}})$
22: $\widehat{\boldsymbol{y}} \leftarrow Softmax(\widetilde{\boldsymbol{y}})$
23: **return** $\widehat{\boldsymbol{c}}, \widehat{\boldsymbol{y}}$

---

### B.2 ENERGY MODEL ARCHITECTURES

Table 2 shows the neural network architectures for different energy functions.

Table 2: Energy Model Architecture.

(a) $\boldsymbol{x}$-$\boldsymbol{y}$ energy network.

| |
|---|
| $\boldsymbol{z}$ = FeatureExtractor($\boldsymbol{x}$) |
| $\boldsymbol{z}$ = FC(Input, hidden) ($\boldsymbol{z}$) |
| $\boldsymbol{z}$ = Dropout(p=0.2) ($\boldsymbol{z}$) |
| $\boldsymbol{u}$ = Embedding($\boldsymbol{y}$) |
| $\boldsymbol{z}$ = $\boldsymbol{z}$ * Norm2($\boldsymbol{u}$) + $\boldsymbol{z}$ |
| $\boldsymbol{z}$ = Relu ($\boldsymbol{z}$) |
| energy = FC(hidden, 1) ($\boldsymbol{z}$) |

(b) $\boldsymbol{x}$-$\boldsymbol{c}$ energy network.

| |
|---|
| $\boldsymbol{z}$ = FeatureExtractor($\boldsymbol{x}$) |
| $\boldsymbol{z}$ = FC(Input, hidden) ($\boldsymbol{z}$) |
| $\boldsymbol{z}$ = Dropout(p=0.2) ($\boldsymbol{z}$) |
| $\boldsymbol{v}$ = Embedding($\boldsymbol{c}$) |
| $\boldsymbol{z}$ = $\boldsymbol{z}$ * Norm2($\boldsymbol{v}$) + $\boldsymbol{z}$ |
| $\boldsymbol{z}$ = Relu ($\boldsymbol{z}$) |
| energy = FC(hidden, 1) ($\boldsymbol{z}$) |

(c) $\boldsymbol{c}$-$\boldsymbol{y}$ energy network.

| |
|---|
| $\boldsymbol{v}$ = Embedding($\boldsymbol{c}$) |
| $\boldsymbol{u}$ = Embedding($\boldsymbol{y}$) |
| cy = $\boldsymbol{u}$ * Norm2($\boldsymbol{v}$) + $\boldsymbol{u}$ |
| cy = Relu (cy) |
| energy = FC(hidden, 1) (cy) |

## C MORE RESULTS

### C.1 VISUALIZATION RESULTS

We selected the ground truth and prediction results of the class "Black and White Warbler" in CUB dataset as a representative case for visualization. We provide the complete 112 concepts heatmaps of joint class-specific of concepts importance ($p(c_{k'} = 1, c_k = 1|\boldsymbol{y})$), class-specific conditional probability among concepts $p(c_k = 1|c_{k'} = 1, \boldsymbol{y})$ and class-agnostic conditional probability among concepts ($p(c_k = 1|c_{k'} = 1)$) in Fig. 5, Fig. 6 and Fig. 7, respectively. Note that $p(c_{k'} = 1, c_k = 1|\boldsymbol{y}) = 1$ leads to $p(c_k = 1|c_{k'} = 1, \boldsymbol{y})$; since most entries ($p(c_{k'} = 1, c_k = 1|\boldsymbol{y})$) in Fig. 4(a) are close to 1, so is Fig. 4(b). Table 3 shows the 20 specific concept names used in Fig. 4.

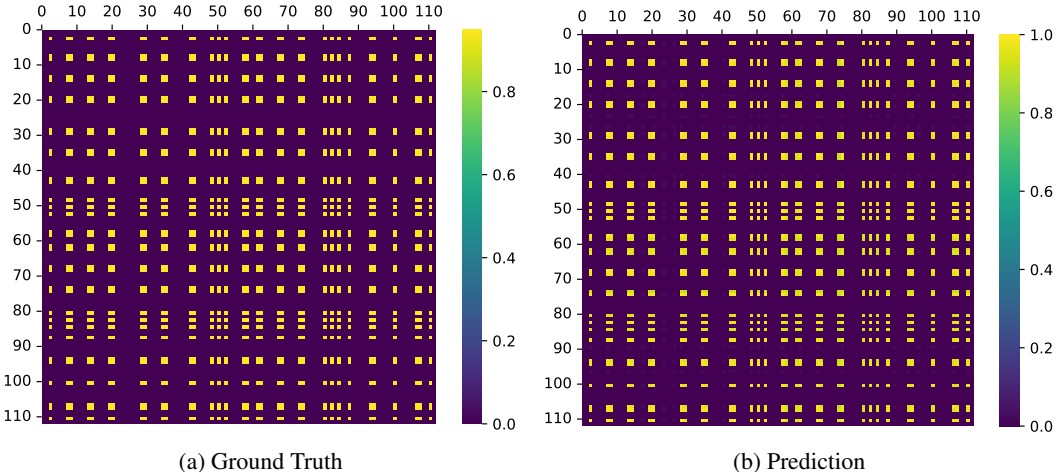

(a) Ground Truth

(b) Prediction

Figure 5: Joint class-specific of concepts importance heatmap ($p(c_{k'} = 1, c_k = 1|\boldsymbol{y})$) for ECBM's ground truth and prediction derived from Proposition 3.3.

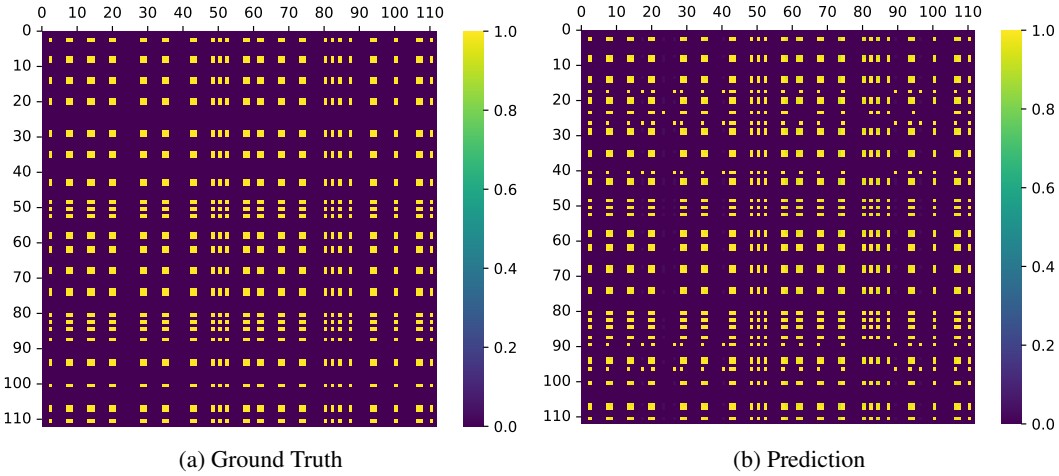

(a) Ground Truth

(b) Prediction

Figure 6: Class-specific conditional probability among concepts heatmap $p(c_k = 1|c_{k'} = 1, \boldsymbol{y})$ for ECBM's ground truth and prediction derived from Proposition 3.4.

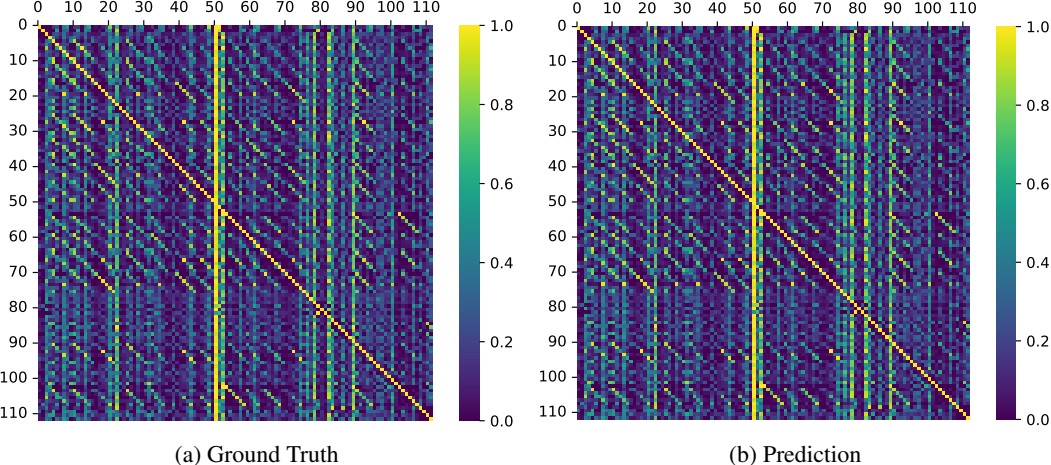

(a) Ground Truth

(b) Prediction

Figure 7: Class-agnostic conditional probability among concepts heatmap ($p(c_k = 1|c_{k'} = 1)$) for ECBM's ground truth and prediction derived from Proposition 3.5.

Table 3: 20 concepts shown in Fig. 4.

| Index | Concept Name |
|---|---|
| 1 | has_bill_shape::curved_(up_or_down) |
| 2 | has_bill_shape::dagger |
| 3 | has_bill_shape::hooked |
| 4 | has_bill_shape::needle |
| 5 | has_bill_shape::hooked_seabird |
| 6 | has_bill_shape::spatulate |
| 7 | has_bill_shape::all-purpose |
| 8 | has_bill_shape::cone |
| 9 | has_bill_shape::specialized |
| 10 | has_wing_color::blue |
| 11 | has_wing_color::brown |
| 12 | has_wing_color::iridescent |
| 13 | has_wing_color::purple |
| 14 | has_wing_color::rufous |
| 15 | has_wing_color::grey |
| 16 | has_wing_color::yellow |
| 17 | has_wing_color::olive |
| 18 | has_wing_color::green |
| 19 | has_wing_color::pink |
| 20 | has_wing_color::orange |

We further provide two baseline results (CBM and CEM) on the complete 112 concepts heatmaps in Fig. 8, Fig. 9 and Fig. 10, respectively.

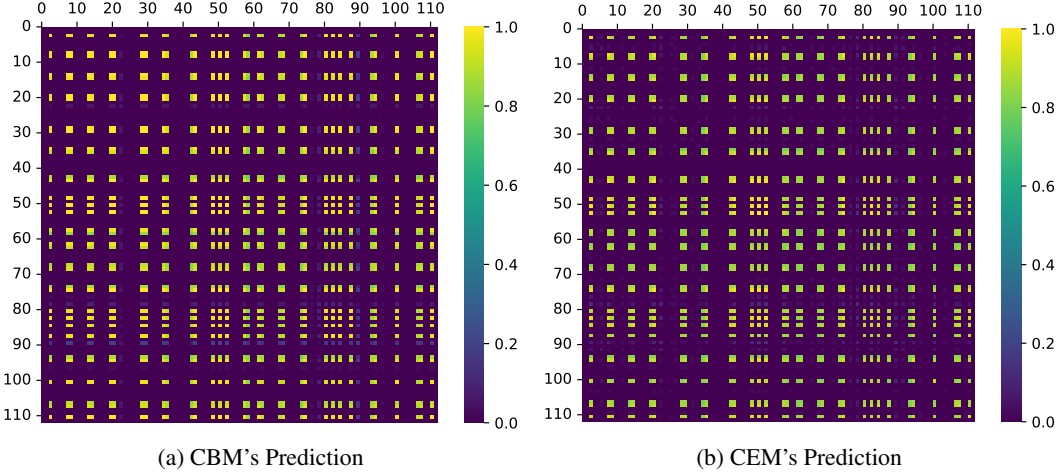

(a) CBM's Prediction         (b) CEM's Prediction

Figure 8: Joint class-specific of concepts importance heatmap ($p(c_{k'} = 1, c_k = 1|\boldsymbol{y})$) for CBM's and CEM's predictions.

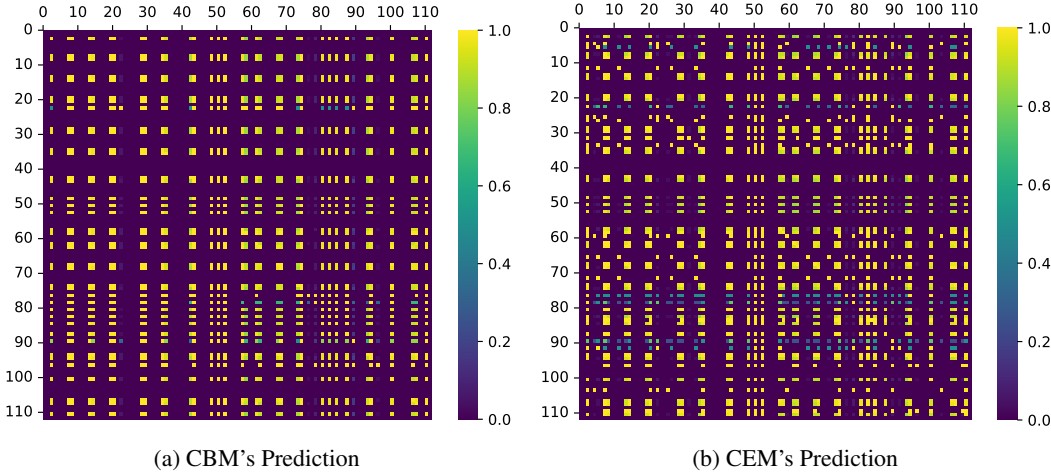

(a) CBM's Prediction        (b) CEM's Prediction

Figure 9: Class-specific conditional probability among concepts heatmap $p(c_k = 1|c_{k'} = 1, \boldsymbol{y})$ for CBM's and CEM's predictions.

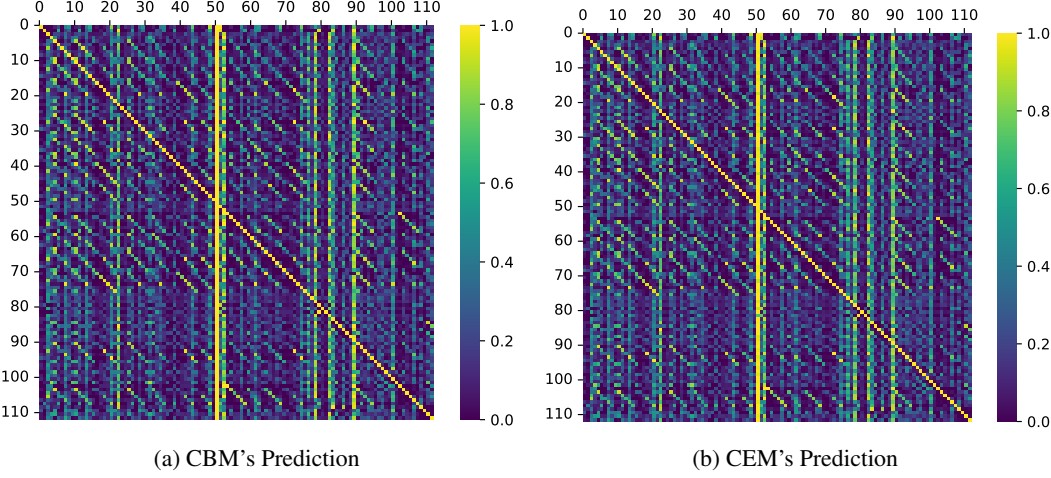

(a) CBM's Prediction        (b) CEM's Prediction

Figure 10: Class-agnostic conditional probability among concepts heatmap ($p(c_k = 1|c_{k'} = 1)$) for CBM's and CEM's predictions.

## C.2 ABLATION STUDY

Table 4 shows the performance of single-branch ECBMs and the black-box model. Table 5 shows the sensitivity of ECBM to its hyperparameters.

Table 4: Ablation study on single-branch ECBMs and the black-box model.

| Data / Model | CUB | | | CelebA | | | AWA2 | | |
|---|---|---|---|---|---|---|---|---|---|
| Metric | Concept | Overall Concept | Class | Concept | Overall Concept | Class | Concept | Overall Concept | Class |
| ECBM ($\boldsymbol{x} - \boldsymbol{y}$ only) | - | - | 0.825 | - | - | 0.265 | - | - | 0.909 |
| ECBM ($\boldsymbol{x} - \boldsymbol{c} - \boldsymbol{y}$ only) | 0.968 | 0.680 | 0.726 | 0.870 | 0.464 | 0.175 | 0.979 | 0.864 | 0.905 |
| **ECBM** | 0.973 | 0.713 | 0.812 | 0.876 | 0.478 | 0.343 | 0.979 | 0.854 | 0.912 |
| CBM | 0.964 | 0.364 | 0.759 | 0.837 | 0.381 | 0.246 | 0.979 | 0.803 | 0.907 |
| CEM | 0.965 | 0.396 | 0.796 | 0.867 | 0.457 | 0.330 | 0.978 | 0.796 | 0.908 |
| Black-box | - | - | 0.826 | - | - | 0.291 | - | - | 0.929 |

Table 5: Hyperparameter Sensitivity Analysis.

| Hyperparameter / Data | CUB | | | CelebA | | | AWA2 | | |
|---|---|---|---|---|---|---|---|---|---|
| Metric | Concept | Overall Concept | Class | Concept | Overall Concept | Class | Concept | Overall Concept | Class |
| $\lambda_l = 0.01$ | 0.971 | 0.679 | 0.756 | 0.872 | 0.456 | 0.166 | 0.979 | 0.854 | 0.907 |
| $\lambda_l = 0.1$ | 0.971 | 0.680 | 0.795 | 0.872 | 0.456 | 0.322 | 0.979 | 0.854 | 0.908 |
| $\lambda_l = 1$ | **0.973** | **0.713** | **0.812** | **0.876** | **0.478** | **0.343** | **0.979** | **0.854** | **0.912** |
| $\lambda_l = 2$ | 0.971 | 0.679 | 0.808 | 0.872 | 0.455 | 0.327 | 0.979 | 0.854 | 0.911 |
| $\lambda_l = 3$ | 0.971 | 0.679 | 0.808 | 0.872 | 0.455 | 0.326 | 0.979 | 0.854 | 0.911 |
| $\lambda_l = 4$ | 0.971 | 0.679 | 0.806 | 0.872 | 0.455 | 0.327 | 0.979 | 0.854 | 0.911 |
| $\lambda_c = 0.01$ | 0.971 | 0.679 | 0.799 | 0.872 | 0.456 | 0.329 | 0.979 | 0.854 | 0.912 |
| $\lambda_c = 0.1$ | 0.971 | 0.679 | 0.799 | 0.872 | 0.456 | 0.166 | 0.979 | 0.864 | 0.912 |
| $\lambda_c = 1$ | **0.973** | **0.713** | **0.812** | **0.876** | **0.478** | **0.343** | **0.979** | **0.854** | **0.912** |
| $\lambda_c = 2$ | 0.971 | 0.679 | 0.798 | 0.872 | 0.455 | 0.329 | 0.979 | 0.854 | 0.912 |
| $\lambda_c = 3$ | 0.971 | 0.679 | 0.798 | 0.872 | 0.455 | 0.329 | 0.979 | 0.854 | 0.912 |
| $\lambda_c = 4$ | 0.971 | 0.679 | 0.798 | 0.872 | 0.455 | 0.329 | 0.979 | 0.854 | 0.912 |
| $\lambda_g = 0.0001$ | 0.971 | 0.679 | 0.805 | 0.872 | 0.455 | 0.326 | 0.979 | 0.854 | 0.911 |
| $\lambda_g = 0.001$ | 0.971 | 0.679 | 0.805 | 0.872 | 0.455 | 0.326 | 0.979 | 0.854 | 0.911 |
| $\lambda_g = 0.01$ | **0.973** | **0.713** | **0.812** | **0.876** | **0.478** | **0.343** | **0.979** | **0.854** | **0.912** |
| $\lambda_g = 0.1$ | 0.971 | 0.680 | 0.795 | 0.872 | 0.456 | 0.322 | 0.979 | 0.854 | 0.909 |
| $\lambda_g = 1$ | 0.971 | 0.680 | 0.756 | 0.872 | 0.456 | 0.166 | 0.979 | 0.854 | 0.907 |

## C.3 ROBUSTNESS

We believe our ECBMs do potentially enjoy stronger robustness to adversarial attacks compared to existing CBM variants. Specifically, our ECBMs are designed to understand the relationships between different concepts, as well as the relationships between concepts and labels. As a result, during inference, ECBMs can leverage these relationships to automatically correct concepts that may be influenced by adversarial attacks. Our preliminary results suggest that our ECBM can potentially improve the robustness against adversarial attacks compared to existing CBM variants.

Table 6: Accuracy on TravelingBirds (Koh et al., 2020) (background shift). We report the CBM results using Table 3 of (Koh et al., 2020).

| Model | Concept | Overall Concept | Class |
|---|---|---|---|
| Standard | - | - | 0.373 |
| Joint (CBM) | 0.931 | - | 0.518 |
| Sequential (CBM) | 0.928 | - | 0.504 |
| Independent (CBM) | 0.928 | - | 0.518 |
| **ECBM** | **0.945** | **0.416** | **0.584** |

Furthermore, we conducted additional experiments on the TravelingBirds dataset following the robustness experiments of CBM (Koh et al., 2020) concerning background shifts. The results (Table 6) reveal that our ECBM significantly outperforms CBMs in this regard. These findings underscore our model's superior robustness to spurious correlations.

## C.4 THE NOTION OF COMPLEX CONDITIONAL RELATIONS

Previous concept-based methods do not allow one to understand "complex" conditional dependencies (as mentioned in the abstract), such as $p(\boldsymbol{c}|\boldsymbol{y})$, $p(c_k|\boldsymbol{y}, c_{k'})$, and $p(\boldsymbol{c}_{-k}, \boldsymbol{y}|\boldsymbol{x}, c_k)$. Post-hoc CBMs and vanilla CBMs, with their interpretative linear layers, provide a way to understand concept-label relationships. In fact, in PCBMs, the weights can indicate each concept's importance for a given label ($p(c_k|\boldsymbol{y})$). However, these existing CBM variants cannot provide more complex conditional dependencies such as $p(c_k|\boldsymbol{y}, c_{k'})$ and $p(\boldsymbol{c}_{-k}, \boldsymbol{y}|\boldsymbol{x}, c_k)$. In contrast, our model, which relies on the Energy-Based Model (EBM) structure, can naturally provide such comprehensive conditional interpretations.

## C.5 INTERVENTION

**Leakage in Models.** We conducted experiments that address the leakage problem as described in (Havasi et al., 2022). When fewer concepts were given in the training process, the model should

not learn as good as the one with all concepts known because of less conceptual information. If the model holds strong performance regardless of concept quantities, it is more likely to suffer from information leakage, i.e., learning concept-irrelevant features directly from image input. This will potentially detriment the accountability of the model interpretation. During training, we provide different quantities of concept groups available for model to learn, and then test the models based on these available concepts. Fig. 11 shows the results. We found that our model performs gradually better when more concept groups are given during training process, instead of intervention-invariant high performances. This indicates that our model suffers less from information leakage, hence providing *more reliable* interpretations.

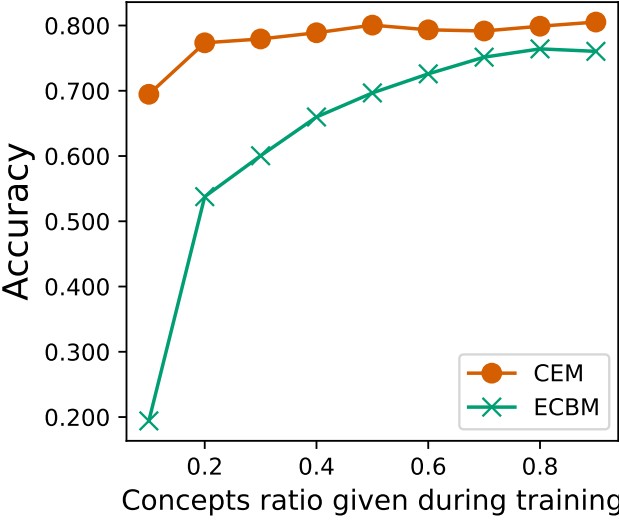

Figure 11: The predictive performance of CEM and hard ECBM on the CUB dataset. The horizontal axis denotes the ratio of concept groups given during the training process, and the vertical axis denotes the "Class Accuracy".

**Individual Intervention on the CUB dataset.** We intervened our model based on individual concepts of the CUB dataset. Fig. 12 shows the intervention results.

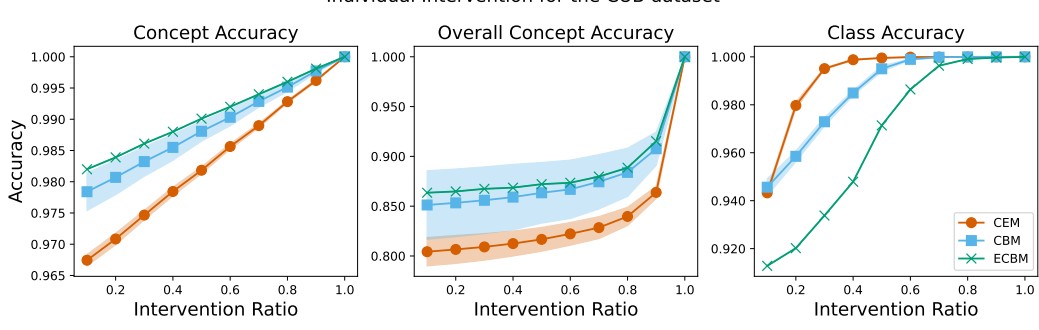

Figure 12: Individual intervention on the CUB dataset (with error bars). The horizontal axis denotes the ratio of the concept group's ground truth given during the inference process.

We have several observations based on these intervention results.

1. Our ECBM underperforms CEM with RandInt in terms of class accuracy. This is expected since CEM is a strong, state-of-the-art baseline with RandInt particularly to improve intervention accuracy. In contrast, our ECBM did not use RandInt. This demonstrates the effectiveness of the RandInt technique that the CEM authors proposed.

2. Even without RandInt, our ECBM can outperform both CBM and CEM in terms of "concept accuracy" and "overall concept accuracy", demonstrating the effective of our ECBM when it comes to concept prediction and interpretation.

3. We would like to reiterate that ECBM's main focus is not to improve class accuracy, but to provide complex conditional interpretation (conditional dependencies) such as $p(\boldsymbol{c}|\boldsymbol{y})$,

$p(c_k|\boldsymbol{y}, c_{k'})$, and $p(\boldsymbol{c}_{-k}, \boldsymbol{y}|\boldsymbol{x}, c_k)$. Therefore, our ECBM is actually complementary to seminal works such as CBM and CEM which focus more on class accuracy and intervention accuracy.

