# OpenReview forum: "Energy-Based Concept Bottleneck Models: Unifying Prediction, Concept Intervention, and Probabilistic Interpretations"
_ICLR.cc/2024/Conference — ICLR 2024 poster_

### Official Review · Reviewer_doCC · 2023-10-30

**Soundness:** 3 good
**Presentation:** 2 fair
**Contribution:** 3 good
**Rating:** 6
**Confidence:** 4

**Summary:**

The paper proposes a new CBM, Energy-Based concept bottleneck model ECBMs.

## Model architecture and training:

Before ECBM, the feature extractor network extracts feature embedding of the input $z$ ECBMs consists of 3 energy networks:
- Class Energy Network: Each label is associated class label embedding $u$, this is inputed to the trainable NN along with $z$, $E_\theta^{class}=G_{zu}(z,u)$. To train the class energy network, the Boltzmann distribution is used to define the conditional likelihood of y given input x. The class energy network maps the input class pair (x, y) to real-valued scalar energy, negative log-likelihood is used as a loss function.
- Concept Energy Network: It consists of K sub-networks (one for each concept), the input is an embedding that represents the aggregate of positive and negative concept embedding $v_k$ (similar to CEM) along with $z$ $E_\theta^{concept}=G_{zv}(z,v_k)$. As done for the class network negative log-likelihood is used as a loss function.

- Global Energy Network: this network learns the dependencies between the concepts and the output. As an input it takes the class embedding  $u$ all concepts embeddings  such that   $E_\theta^{global}=G_{vu}([v_{k}]_{k=1}^{K},u)$. The negative log-likelihood loss is calculated for each concept-class pair.


## Prediction:
To make a prediction feature, the feature network and energy networks are frozen then search for optimal prediction of concepts $\hat{c}$ and the class label $\hat{y}$ by minimizing equation 12.

## Interventions:
One of the main advantages of ECBMs over other methods is that they are able to propagate the corrected concept to other correlated
concepts, this is done automatically while optimizing for $\hat{c}$ and $\hat{y}$ and changing some values in $\hat{c}$ to ground-truth $c$.

## Conditional interpretations:
By using the energy functions given a target class one can calculate the marginal concept importance for that class, the joint concepts importance for that class, the conditional probability between concepts for that class and the conditional probability between concepts (without taking class into consideration i.e the conditional probability summed over classes).

## Experiments:
- The paper compared ECBMs with CBMs, CEMs, PCBMs and ProbCBM; across 3 datasets.
- For evaluation, the paper compared (a) Concept accuracy (i.e. predict each concept independently). (B) The overall concept accuracy, this evaluates that all concepts are predicted correctly for a given sample i.e concept vector == ground-truth vector. (C) The class accuracy.
- For the experiments they looked at concept and class label prediction (without any interventions), concept and class label prediction with interventions and conditional interpretation.

**Strengths:**

## Originality-Excellent:
- The idea to use energy-based models for concepts is quite original.
- The training and inference algorithm is also original and the idea has not been explored before.

## Novelty-Excellent:
Besides the model architecture and loss being novel there is two additional aspects that are also novel about this work:
- As far as I know, this is the first work that captures concept interaction during intervention.
- The conditional interpretations have not been investigated previously.

## Quality-Good:
- Based on the result section, for  ECBMs outperform baselines for concept and class label prediction experiments. For the interventions, ECBMs seem to outperform baselines in concept accuracy and most interestingly the overall concept accuracy, which supports the intilal claim of the paper that by correcting one class will help other classes can be corrected as well. However, CBMs outperform in class accuracy which is somewhat expected.

**Weaknesses:**

## Clarity -- Poor:
- In page 1, under limitations of CBMs the paper mentioned interpretability and said "They cannot effectively quantify the intricate relationships between various concepts and class labels". I disagree, in the original CBM on top of the CBM layer there is a linear which quantifies how each concept affects the final output so if a model classifies an image with label "A" we can look at the weight matrix and see which concepts where used in this classification yes at inference time (assuming we don't have the ground truth concepts) we don't know if the model correctly classified the concepts or not or the likelihood of a concept being correctly classified but we do know the relationship between the concepts and the final output.
- The paper is not very clear, please see the questions section.

## Significance --Fair:
- ECBMs Training: Training is much more complicated than regular CBMs and probably much more expensive particularly optimizing equation 10.
- ECBMs Inference requires optimization not just a forward pass through the network.
- Usage and reproducibility: There was no code to reproduce the experiment. It is very difficult to imagine how all of this is trained together without the code.

**Questions:**

- What is the accuracy of a black-box model for datasets in table 1?
- Results in figure 2 are from which dataset?
- Aren't $E_\theta^{class}$, $E_\theta^{concept}$,$E_\theta^{global}$ neural networks? If so why don't they have trainable or frozen signs in figure 1?
- Equation 9 is $[v_k]_{k=1}^{K}$ this the concatenation of all concepts embeddings?
- Please clearly state the output dimensions of each energy model.
- For the gradient-based inference Algorithm 1 line 14,15,16 since this is done during inference the algorithm shouldn't have access to $c,y$ ? Is this a typo?

---

> ### Author Response · Authors · 2023-11-20
> **[1/2] Thank you for the constructive and encouraging comments.**
>
> Thank you for your valuable reviews. We are glad that you acknowledged that our ECBM is ``"the first work that captures concept interaction during intervention"``, and found our novelty ``"excellent"`` and our idea ```"original"```. Below we address your questions one by one.
>
> For your general questions on clarity, significance, and code release:
>
> - **Clarity: Final Linear Layer of CBM as Conditional Interpretation.** We apologize if our presentation has led to confusion. By 'intricate', we were referring to complex conditional dependencies such as $p(c|y)$, $p(c_{k}|y,c_{k^{\prime}})$, and $p(c_{-k}, y|x,c_{k})$. We agree that Post-hoc CBMs and vanilla CBMs, with their interpretative linear layers, provide a way to understand concept-label relationships. In fact, in PCBMs, the weights can indicate each concept's importance for a given label ($p(c_k|y)$).
> However, these existing CBM variants cannot provide more complex conditional dependencies such as $p(c_{k}|y,c_{k^{\prime}})$ and $p(c_{-k}, y|x,c_{k})$. In contrast, our model, which relies on the Energy-Based Model (EBM) structure, can naturally provide such comprehensive conditional interpretations.
> - **Significance: Training Cost.** Training the ECBMs is indeed more complex than training regular CBMs. To manage the training cost, we employ a negative sampling strategy, similar to traditional CBMs' cost.
> - **Significance: Inference Cost.** Indeed, due to the energy-based nature of our model, our ECBM's inference requires multiple optimization iterations rather than a single feedforward pass. There are strategies to expedite this inference process. One such approach is to use the predictions from a vanilla CBM as an initialization, thereby 'jump-starting' the optimization and substantially accelerating inference.
> - **Usage and Reproducibility: Code Release.** Thank you for your encouraging comments and interest in our work. We understand the importance of reproducibility in scientific research. So far, we have finished cleaning up the source code and will release it if the paper is accepted.
>
> ## Questions
>
> **Q1. What is the accuracy of a black-box model for datasets in table 1?**
>
> Thank you for mentioning this. For CBMs, the standard practice is to take the vanilla CBM as the baseline. However, in response to your suggestion and for the sake of a more comprehensive evaluation, we have also tested black-box models and included the results in our revised Table 4 in Appendix C. These results show that
>
> + our ECBM achieves performance very close to the black-box model on the CUB and AWA2 datasets; for example, on CUB, the accuracy of the black-box model and our ECBM is $0.826$ and $0.812$, respectively.
> + In some cases, such as with the CelebA dataset, it even improves upon the black-box performance ($0.343$ versus $0.291$). This can be attributed to our joint inference process of concepts and class labels.
> + our ECBM outperforms all baseline models, including CBM and CEM.
>
> **Q2. Results in Figure 2 are from which dataset?**
>
> Our apologies for the oversight. The results displayed in Figure 2 are from the CUB dataset. We have updated the figure caption in the revised manuscript to include this information. Thank you for pointing this out.
>
> **Q3. Aren't $E_{\theta}^{class}$, $E_{\theta}^{concept}$, $E_{\theta}^{global}$ neural networks? If so why don't they have trainable or frozen signs in figure 1?**
>
> Thanks for mentioning this. Indeed, $E_{\theta}^{class}$, $E_{\theta}^{concept}$, and $E_{\theta}^{global}$ are neural networks. The absence of trainable or frozen signs in Figure 1 was purely for aesthetic and clarity purposes. We apologize for any confusion this may have caused. According to your suggestion, we have updated Figure 1 to include these signs.
>
> **Q4. Equation 9 is $[v_k]_{k=1}^{k}$ this the concatenation of all concepts embeddings?**
>
> Yes. In Equation 9, $[v_k]_{k=1}^{k}$ denotes the concatenation of all concept embeddings. Thank you for pointing this out, and we have clarified this in our revision accordingly.
>
> **Q5. Please clearly state the output dimensions of each energy model.**
>
> We are sorry for the confusion. Each energy model (e.g., $E_{\theta}^{concept}$) outputs *a scalar value*, which measures the compatibility between the variables. We hope this addresses your question. If you require further clarification or have additional questions, we will be more than happy to follow up with any details needed.

---

> ### Author Response · Authors · 2023-11-20
> **[2/2] Thank you for the constructive and encouraging comments.**
>
> **Q6. For the gradient-based inference Algorithm 1 line 14,15,16 since this is done during inference the algorithm shouldn't have access to $\mathbf{c}, \mathbf{y}$ ? Is this a typo?**
>
> We are sorry for the typo. During inference, the algorithm does not have access to $c, y$. In the revision, we have corrected this by changing them to $\widehat{c}, \widehat{y}$ (i.e., the estimated concepts and class labels in the current iteration) in the algorithm.

---

> ### Comment · Reviewer_doCC · 2023-11-21
> **Thanks for the clarification**
>
> I would like to thank the authors for the clarifications. The changes made in the manuscript made it much clearer! My acceptance recommendation remains unchanged.

---

### Official Review · Reviewer_WpSc · 2023-10-31

**Soundness:** 1 poor
**Presentation:** 2 fair
**Contribution:** 2 fair
**Rating:** 8
**Confidence:** 4

**Summary:**

This paper proposes Energy-based Concept Bottleneck Models (ECBMs), a novel concept-based architecture for predicting task labels and providing concept-based explanations for those predictions. The proposed model addresses (1) the lack of concept-based models that can capture relationships between concepts and labels and (2) the inability of test-time concept interventions in existing concept-based models to propagate to concepts besides those being intervened on. ECBMs achieve these goals via the use of energy functions that capture the joint probabilities between input features, concepts, and task labels. By learning these functions in an end-to-end fashion with an aggregate loss function, and composing these energy functions at inference time, one can (1) produce task predictions with corresponding concept explanations, (2) estimate concept importance for specific labels, and (3) impute mispredicted unintervened concepts after an intervention has been performed, therefore improving the impact of interventions on a model's performance. This paper evaluates ECBMs on three real-world datasets and shows that they can outperform existing baselines on both task and concept accuracy.

**Strengths:**

Thank you for submitting this work. I found the general main idea to be very interesting and worth exploring. In particular, I believe that the following are the main strengths of this paper:

1. **Originality**: the idea of incorporating energy functions to model inputs, concepts, and task labels concurrently is, to the best of my knowledge, something that has not been attempted in previous work in the concept-based literature. In particular, I found the use of these energy functions to propagate interventions to be both original and interesting.
2. **Quality**: although the quality of the experiments could be significantly improved on (see below), the quality of the presented idea, as the presentation of the idea itself, meets the standard of what I would expect in this field and is very clearly elaborated.
3. **Clarity**: the paper’s writing is extremely clear and easy to follow. This helps the authors clearly communicate their ideas and the motivation behind the ideas. Although this work could benefit from further details when it comes to elements crucial to reproducibility (see below), I deeply thank the authors for writing in a very concise and clear manner throughout the manuscript. This lack of details, however, is the reason behind my "fair" score in the presentation field above.
4. **Significance**: the paper’s main contribution, that of reframing concept learning as learning sets of joint probability distributions, can lead to novel and significant ideas in the future. It is certainly a novel and significant way to think about concept learning. Nevertheless, this significance is contingent on the paper’s core idea being carefully proven to work as expected (see weaknesses below).  If the experiments are carefully corrected for the weaknesses stated below, and the paper’s main claims still hold, then I believe this paper would be of good value to the community.

**Weaknesses:**

Although I found several parts of this work to be strong and well-defined, I have the following hesitations regarding this paper in its current state:

1. [Critical] In my opinion, the biggest weakness of this paper is its evaluation and the fairness of the evaluation itself. As discussed in the questions below, this paper seems to compare baselines with significantly varying capacities and fine-tuning processes ( e.g., ProbCBM is trained with a ResNet18 backbone in the source of the results reported in this paper vs ECBM trained with a ResNet101 backbone) as well as failing to evaluate key contributions across several tasks (e.g., interventions are evaluated on a single task). This yields a lot of the evaluation inconclusive as it makes the source of the improvements shown in the proposed model hard to identify.
2. [Critical] Against common practices in the field, none of the results in the evaluation section include error bars. Without such error bars, it is very hard to judge for statistical significance. This is particularly important for the paper’s main evaluation in Table 1 where differences between ECBMs and other baselines are small for task (e.g., ”class”) accuracies on some tasks.
3. [Critical] Some of the key baseline results in this paper seem to contradict the results presented in previous literature. For example, CEM’s intervention performance in Figure 2 is in direct contradiction to those reported in the original paper and in subsequent works ([1, 2, 3]). Furthermore, the authors claim that ProbCBMs cannot be trained on CelebA due to the binary nature of the concept annotations. This however, to the best of my understanding, is not a requirement for ProbCBM (an example is CUB itself which is used as a baseline for this method even though it has not uncertainty labels as claimed by the manuscript). Similarly, results for Post-hoc CBMs are missing for CelebA and AWA2 as the authors claim that Post-hoc CBMs require "predefined concept activation vectors." This is not true as the CAVs needed for PCBMs can be easily learnt from a dataset with binary concept annotations as those in CelebA and AWA2. In fact, the authors of that paper do this exact process for CUB.
4. [Critical] A lot of details required for reproducibility are missing. There is no code attached to this submission as part of the supplement. There are no details on the details and training procedures of any of the baselines. The hyperparameter selection is not discussed anywhere, which brings doubt to the very specific precision indicated for some of ECBM’s selected hyperparameters (e.g., $\lambda_c = 0.49$).
5. [Major] Related to the point above, there are no ablation studies showing the sensitivity of the proposed method to its hyperparameters ($\lambda_{\{c, g\}}$).
6. [Major] The claim that previous concept-based methods do not allow one to understand concept to label relationships is not entirely true. Post-hoc CBMs enable this via their sparse interpretable predictor and even vanilla CBMs tend to be trained with a linear layer on top of their concept bottleneck, rendering an easy mechanism to obtain concept importance for a given label via the linear layer's weights.
7. [Major] The proposed evaluation is limited in the number and quality of the competing baselines. For example, several of the baselines are missing values in Table 1.
8. [Minor] Inference on ECBMs requires one to solve an optimization problem via a relaxation that is solved with gradient descent + back-propagation. This means that this proposed approach may not scale well to a setup where one is expected to run inference quickly or as part of a separate pipeline, which needs to be end-to-end differentiable.
9. [Minor] The paper lacks any discussion of the limitations of its method. The last section, which has “limitations” on its title, fails to discuss any of the limitations.

These weaknesses, in my opinion, lead to significant doubt on the soundness of the discussed results and therefore contribute to my decision leading towards rejection. Nevertheless, if I misunderstood any significant pieces of this work, or if some of these weaknesses are void, I am happy to be convinced otherwise if the appropriate evidence is provided.

### References

[1] Sheth, Ivaxi, and Samira Ebrahimi Kahou. "Overcoming Interpretability and Accuracy Trade-off in Medical Imaging." *Medical Imaging with Deep Learning, short paper track*. 2023.

[2] Collins, Katherine Maeve, et al. "Human uncertainty in concept-based ai systems." *Proceedings of the 2023 AAAI/ACM Conference on AI, Ethics, and Society*. 2023.

[3] Zarlenga, Mateo Espinosa, et al. "Learning to Receive Help: Intervention-Aware Concept Embedding Models." *arXiv preprint arXiv:2309.16928* (2023).

**Questions:**

The following questions, in no particular order, would help clarify some of the doubts on this work. If possible, I would appreciate it if the authors could elaborate on these concerns as they may serve as a good starting point for a discussion during rebuttal:

1. Could you please elaborate on the seemingly contradicting baseline results discussed in the weaknesses? In particular, for the CEM interventions, the shown results seem to match the interventions in CEMs when `RandInt` is not used for training (something the authors explicitly discuss is necessary for interventions to work properly). Is it the case that you trained CEMs with RandInt? I tried to look for code as part of this submission but could not find it in the supplement (please let me know if I missed it; in which case, I am very sorry for that!).
2. Related to the question above, which dataset is used in the evaluation of Figure 2? Is this CUB? I could not find this information anywhere. Also, when intervening on a bottleneck, are interventions performed on groups of concepts or on individual concepts at a time? In CUB, in particular, most previous work focuses on group-level interventions, so clarifying this is very important. Based on the left-most plot in Figure 1 increasing linearly, I am assuming interventions are done at an individual level rather than at a group level, but clarifying this is very important.
3. Similarly, could you please elaborate on how ECBMs fare against competing baselines when intervened on in the other datasets?
4. Could you please elaborate on the lack of results for PCBMs and ProbCBMs given my comments on the weaknesses indicating that they are in fact baselines that could be evaluated in the setups used in this paper?
5. Could you please elaborate more on the specifics of the negative sampling strategy used to learn $E^\text{global}_\theta$? This is a critical practical element of this method and including details for this is crucial if you want to ensure reproducibility. Furthermore, understanding how this is done to scale for large concept spaces is extremely important to understand some of the limitations of this work.
6. The outer summation of Preposition 3.2 is very unclear. It seems to be “iterating” over all concept vectors without considering the $k$-th dimension, but it is not immediately clear if this is the case or whether these vectors are what $\mathbf{c}$ is referred to inside the summation. Can you please clarify what are this summation’s bounds and iteration variables?
7. When learning the positive and negative embeddings for each concept, I was wondering if you have an intuition as to what inductive bias in your model is responsible for forcing the distribution of positive and negative embeddings to be easily separable? For example, in ProbCBMs the use of a contrastive loss with two “anchor” embeddings leads to this separation. Similarly, in CEMs the use of RandInt at training-time leads to this separation.  Understanding this is crucial for understanding why a given model performs well or badly when intervened on and may help clarify the design choice/process of this architecture to future readers hoping to build on top of this work.

Besides these questions, I found the following typos/minor errors that, if fixed, could improve the quality of the current manuscript:

1. In section 3.2 (page 5), “the second terms” should be “the second term”
2. In section 3.3 (page 6), ”Proposition 3.2 below compute” should be “Proposition 3.2 below computes”
3. In proposition 3.2, the variable $k$ is already captured when using $c_k$ but it is then overwritten in the summation over all concepts in the RHS’s numerator that uses $k$ again to indicate the iteration index. The same applies for propositions 3.4 and 3.5!
4. In section 3.4 (page 6), “Proposition 3.3 below compute” should be “Proposition 3.3 below computes”

---

> ### Author Response · Authors · 2023-11-20
> **[1/5] Thank you for the constructive and encouraging comments.**
>
> Thank you for your insightful and constructive feedback. We are glad that you found the problem we sovle ```"novel and significant"```/```"interesting"```, our paper ```"easy to follow"```/```"clear"```.  Below we address your questions one by one.
>
>
>
> **W1.  ...In my opinion, the biggest weakness of this paper is its evaluation and the fairness of the evaluation itself...ResNet18...ResNet101...**
>
> Thank you for mentioning this.
> Our preliminary results show that ProbCBM with ResNet101 underperforms ProbCBM with ResNet18. We suspect that probCBM may exhibit instability as the dimensionality of the feature embedding increases. Hence we have chosen to report the highest scores in the paper. The table below shows the results.
>
> |       Dataset       |         |       CUB       |       |         |     CelebA      |       |         |      AWA2       |       |
> | :-----------------: | :-----: | :-------------: | :---: | :-----: | :-------------: | :---: | :-----: | :-------------: | :---: |
> |       Metric        | Concept | Overall Concept | Class | Concept | Overall Concept | Class | Concept | Overall Concept | Class |
> | ProbCBM (ResNet18)  |  0.946  |      0.360      | 0.718 |    -    |        -        |   -   |  0.959  |      0.719      | 0.880 |
> | ProbCBM (ResNet101) |  0.932  |      0.236      | 0.650 |  0.888  |      0.511      | 0.330 |  0.930  |      0.553      | 0.739 |
> |      **ECBM**       |  0.973  |      0.713      | 0.812 |  0.876  |      0.478      | 0.343 |  0.979  |      0.854      | 0.912 |
>
>
> Furthermore, we expanded our evaluation to include more intervention experiments and adopted interventions across all datasets. This approach allows us to demonstrate the intervention performance under varying ratios (more details in the response to **Q1-Q3** below). We believe your suggestions have significantly enhanced the comprehensiveness of our experiment. Thank you for your valuable input.
>
> **W2. Against common practices in the field, none of the results in the evaluation section include error bars.**
>
> This is a good suggestion. Accordingly, we have revised Table 1 to include standard deviations over five runs with different random seeds. These will help present a clearer perspective on the statistical significance of our results. Our standard deviations range from 0.000 to 0.006, except for the overall concept accuracy for CUB, which is 0.009. All values are represented up to three decimal digits. These results confirm the significance of the improvement brought by our ECBMs.
>
> **W3. Some of the key baseline results in this paper seem to contradict the results presented in previous literature.**
>
> We are sorry for the confusion. It seems the inconsistency with prior intervention performance might be due to our use of individual-level concept intervention. In light of your feedback, we re-conducted the intervention experiment using group-level intervention (more details in the response to **Q1-Q3** below).
>
> For the results of ProbCBM and Post-hoc CBM (PCBM), we acknowledge the misunderstandings regarding their requirements and capabilities. We revisited these models and retrained them accordingly. These revised results are now included in Table 1 of our revised paper. Your suggestions and guidance have been invaluable in addressing these issues. Thank you for drawing our attention to these points. The new results confirm that our ECBM outperforms these baselines (ProbCBM and PCBM) in both CeleA and AWA2 too.
>
>
> **W4. A lot of details required for reproducibility are missing.**
>
> Thank you for mentioning this. We understand the importance of transparency in research and are committed to making our work as accessible and replicable as possible.
>
> For the code release, so far, we have finished cleaning up the source code and will release it if the paper is accepted.
>
> For the baselines, we assure you that we have strictly adhered to the original papers and their provided codebases for reproduction.
>
> Regarding hyperparameter $\lambda_{c}=0.49$, this is because in our implementation, we introduce $\lambda_l$ to Eqn. (12) in the paper, resulting in:
> $\lambda_l E_{\theta}^{class}(x,y) + \lambda_c E_{\theta}^{concept}(x,c) + \lambda_g E_{\theta}^{global}(c,y)$
>
> We then set $\lambda_l=1$, $\lambda_c=1$ and $\lambda_g=0.01$.
>
> This is therefore equivalent to setting $\lambda_c=1/(1+1+0.01)=0.49$ for the original Eqn. (12) without $\lambda_l$:
> $E_{\theta}^{class}(x,y) + \lambda_c E_{\theta}^{concept}(x,c) + \lambda_g E_{\theta}^{global}(c,y)$.
>
> We are grateful for your attention to these details and have revised the paper accordingly.

---

> ### Author Response · Authors · 2023-11-20
> **[2/5] Thank you for the constructive and encouraging comments.**
>
> **W5. Related to the point above, there are no ablation studies showing the sensitivity of the proposed method to its hyperparameters.**
>
> In response to your feedback, we have conducted ablation studies to demonstrate the sensitivity of our proposed method to its hyperparameters. The results on the CUB dataset are available in the table below. These studies provide insights into the hyperparameter search process and show that our ECBM is stable over a wide range of hyperparameters. We appreciate your valuable suggestion and have included all additional results on CUB, AWA2, and CelebA datasets in our revised paper (Table 5 of Appendix C.2).
>
> |   Hyperparameter   | Concept | Overall Concept | Class |
> | :----------------: | :-----: | :-------------: | :---: |
> |  $\lambda_l=0.01$  |  0.971  |      0.679      | 0.756 |
> |  $\lambda_l=0.1$   |  0.971  |      0.680      | 0.795 |
> |   $\lambda_l=1$    |  0.973  |      0.713      | 0.812 |
> |   $\lambda_l=2$    |  0.971  |      0.679      | 0.808 |
> |   $\lambda_l=3$    |  0.971  |      0.679      | 0.808 |
> |   $\lambda_l=4$    |  0.971  |      0.679      | 0.806 |
> |  $\lambda_c=0.01$  |  0.971  |      0.679      | 0.799 |
> |  $\lambda_c=0.1$   |  0.971  |      0.679      | 0.799 |
> |   $\lambda_c=1$    |  0.973  |      0.713      | 0.812 |
> |   $\lambda_c=2$    |  0.971  |      0.679      | 0.798 |
> |   $\lambda_c=3$    |  0.971  |      0.679      | 0.798 |
> |   $\lambda_c=4$    |  0.971  |      0.679      | 0.798 |
> | $\lambda_g=0.0001$ |  0.971  |      0.679      | 0.805 |
> | $\lambda_g=0.001$  |  0.971  |      0.679      | 0.805 |
> |  $\lambda_g=0.01$  |  0.973  |      0.713      | 0.812 |
> |  $\lambda_g=0.1$   |  0.971  |      0.680      | 0.795 |
> |   $\lambda_g=1$    |  0.971  |      0.680      | 0.756 |
>
> **W6. The claim that previous concept-based methods do not allow one to understand concept to label relationships is not entirely true.**
>
>
> We apologize if our presentation has led to confusion. We meant to say that previous concept-based methods do not allow one to understand ``"complex"`` conditional dependencies (as mentioned in the abstract) such as $p(c|y)$, $p(c_{k}|y,c_{k^{\prime}})$, and $p(c_{-k}, y|x,c_{k})$. We agree that Post-hoc CBMs and vanilla CBMs, with their interpretative linear layers, provide a way to understand concept-label relationships. In fact, in PCBMs, the weights can indicate each concept's importance for a given label ($p(c_k|y)$). However, these existing CBM variants cannot provide more complex conditional dependencies such as $p(c_{k}|y,c_{k^{\prime}})$ and $p(c_{-k}, y|x,c_{k})$. In contrast, our model, which relies on the Energy-Based Model (EBM) structure, can naturally provide such comprehensive conditional interpretations. Thank you for pointing this out, and we have clarified this in the revision as suggested.
>
>
> **W7. The proposed evaluation is limited in the number and quality of the competing baselines.**
>
> Thank you for mentioning this. According to your feedback, we have updated Table 1 to provide a more comprehensive and fair comparison (see more discussion in the response to **W1-W3** above). We appreciate your suggestion, which has helped improve the quality of our evaluation.
>
> **W8. Inference on ECBMs requires one to solve an optimization problem via a relaxation that is solved with gradient descent + back-propagation.**
>
> Indeed, due to the energy-based nature of our model, our ECBM's inference requires multiple optimization iterations rather than a single feedforward pass. There are strategies to expedite this inference process. One such approach is to use the predictions from a vanilla CBM as an initialization, thereby 'jump-starting' the optimization and substantially accelerating inference. We believe further research into accelerating energy-based CBMs would be interesting future work. We have included the discussion above in the revision.
>
> **W9. The paper lacks any discussion of the limitations of its method.**
>
> We appreciate your feedback regarding the omission of a discussion on the limitations of our method in the last section of the paper. We are sorry for the oversight and agree that such a discussion is crucial for a balanced presentation of our work. In response to your observation, we will add a comprehensive discussion of the limitations of ECBMs to the final section.
>
> Your feedback has helped us identify several potential limitations, including the need for better solutions in the optimization process of our ECBM and the design of efficient initialization procedures. Additionally, our current model requires concept labeling, a process that could be improved with label-free approaches. As a potential future improvement, our model could be adapted to work with label-free CBMs. Thank you for your constructive feedback, it has been instrumental in helping us improve our work.

---

> ### Author Response · Authors · 2023-11-20
> **[3/5] Thank you for the constructive and encouraging comments.**
>
> ## Questions
>
> **Q1.1. Could you please elaborate on the seemingly contradicting baseline results discussed in the weaknesses?**
>
> Thank you for mentioning this. Indeed, in our initial experiments with the CEM interventions, we used the vanilla CEM without RandInt for training. In response to your feedback, we have supplemented the experiments where CEMs are trained *with RandInt*. The new results can be found in Figure 12 of Appendix C.5. This should provide a more accurate reflection of the CEM intervention performance.
>
>
> **(1)** Our ECBM underperforms CEM *with RandInt* in terms of class accuracy. This is expected since CEM is a strong, state-of-the-art baseline *with RandInt* particularly to improve intervention accuracy. In contrast, our ECBM did not use RandInt. This demonstrates the effectiveness of the RandInt technique that the CEM authors proposed; we are in the process of incorporating it into our ECBM to see whether it would improve ECBM's class accuracy. We will update the results in the author feedback if we could make it before the discussion deadline (Nov 22).
>
> **(2)** Even *without RandInt*, our ECBM can outperform both CBM and CEM in terms of "concept accuracy" and "overall concept accuracy", demonstrating the effectiveness of our ECBM when it comes to concept prediction and interpretation.
>
> **(3)** We would like to reiterate that ECBM's main focus is not to improve class accuracy, but to provide *"complex"* conditional interpretation (conditional dependencies) such as $p(c|y)$, $p(c_{k}|y,c_{k^{\prime}})$, and $p(c_{-k}, y|x,c_{k})$. (Please see the response to **W6** for more details on the definition of *"complex"*.) Therefore, our ECBM is actually complementary to seminal works such as CBM and CEM which focus more on class accuracy and intervention accuracy.
>
> We have included the discussion above in our revision as suggested.
>
>
> **Q1.2. ...look for code...**
>
> Thank you for your interest in our work. We understand the importance of transparency in research and are committed to making our work as accessible and replicable as possible. For the code release, so far, we have finished cleaning up the source code and will release it if the paper is accepted.
>
>
> **Q2.1. Related to the question above, which dataset is used in the evaluation of Figure 2?**
>
> Our apologies for the oversight. The results displayed in Figure 2 are from the CUB dataset. We have updated the figure caption in the revised manuscript to include this information. Thank you for pointing this out.
>
>
> **Q2.2. ...are interventions performed on groups of concepts or on individual concepts at a time?...**
>
> Thank you for pointing this out. Indeed, we performed interventions on individual concepts. Following your suggestion, we have also added group-level interventions and updated our results accordingly in Figure 12 of Appendix C.5. This additional data provides a more comprehensive view of our model's efficacy in different intervention scenarios. Please also see the response to **Q1.1** above for details observations from the new results in Figure 12.

---

> ### Author Response · Authors · 2023-11-20
> **[4/5] Thank you for the constructive and encouraging comments.**
>
> **Q3. Similarly, could you please elaborate on how ECBMs fare against competing baselines when intervened on in the other datasets?**
>
> This is a good suggestion. We have followed your suggestion to include results on concept intervention for other datasets, i.e., CeleA and AWA2. The results have been incorporated into our revised paper and can be found in Figure 13 and 14 of Appendix C.5. Note that CeleA and AWA2 do not have *group assignment*, therefore we perform individual-level intervention for these two datasets. Similar to the results for CUB, we have the following observations on how ECBMs fare against baselines:
>
>
> **(1)** Our ECBM underperforms CEM *with RandInt* in terms of class accuracy. This is expected since CEM is a strong, state-of-the-art baseline *with RandInt* particularly to improve intervention accuracy. In contrast, our ECBM did not use RandInt. This demonstrates the effectiveness of the RandInt technique that the CEM authors proposed; we are in the process of incorporating it into our ECBM to see whether it would improve ECBM's class accuracy. We will update the results in the author feedback if we could make it before the discussion deadline (Nov 22).
>
> **(2)** Even *without RandInt*, our ECBM can outperform both CBM and CEM in terms of "concept accuracy" and "overall concept accuracy", demonstrating the effectiveness of our ECBM when it comes to concept prediction and interpretation.
>
> **(3)** We would like to reiterate that ECBM's main focus is not to improve class accuracy, but to provide *"complex"* conditional interpretation (conditional dependencies) such as $p(c|y)$, $p(c_{k}|y,c_{k^{\prime}})$, and $p(c_{-k}, y|x,c_{k})$. (Please see the response to **W6** for more details on the definition of *"complex"*.) Therefore, our ECBM is actually complementary to seminal works such as CBM and CEM which focus more on class accuracy and intervention accuracy.
>
> **Q4. Could you please elaborate on the lack of results for PCBMs and ProbCBMs given my comments on the weaknesses indicating that they are in fact baselines that could be evaluated in the setups used in this paper?**
>
> Thank you for your insightful comments. We acknowledge the misunderstandings regarding their requirements and capabilities. We revisited these models (PCBMs and ProbCBMs) and retrained them accordingly. The table below shows the revised results and is now incorporated into the revised Table 1 of our main paper.
>
> |  Data   |        CUB        |                   |                   |      CelebA       |                   |                   |       AWA2        |                   |                   |
> | :-----: | :---------------: | :---------------: | :---------------: | :---------------: | :---------------: | :---------------: | :---------------: | :---------------: | :---------------: |
> | Metric  |      Concept      |  Overall Concept  |       Class       |      Concept      |  Overall Concept  |       Class       |      Concept      |  Overall Concept  |       Class       |
> |   CBM   | 0.964$ \pm$ 0.002 | 0.364 $\pm$ 0.070 | 0.759 $\pm$ 0.007 | 0.837 $\pm$ 0.009 | 0.381 $\pm$ 0.006 | 0.246 $\pm$ 0.005 | 0.979 $\pm$ 0.002 | 0.803 $\pm$ 0.023 | 0.907 $\pm$ 0.004 |
> | ProbCBM | 0.946 $\pm$ 0.001 | 0.360 $\pm$ 0.002 | 0.718 $\pm$ 0.005 | 0.867 $\pm$ 0.007 | 0.473 $\pm$ 0.001 | 0.299 $\pm$ 0.001 | 0.959 $\pm$ 0.000 | 0.719 $\pm$ 0.001 | 0.880 $\pm$ 0.001 |
> |  PCBM   |         -         |         -         | 0.635 $\pm$ 0.002 |         -         |         -         | 0.150 $\pm$ 0.010 |         -         |         -         | 0.862 $\pm$ 0.003 |
> |   CEM   | 0.965 $\pm$ 0.002 | 0.396 $\pm$ 0.052 | 0.796 $\pm$ 0.004 | 0.867 $\pm$ 0.001 | 0.457 $\pm$ 0.005 | 0.330 $\pm$ 0.003 | 0.978 $\pm$ 0.008 | 0.796 $\pm$ 0.011 | 0.908 $\pm$ 0.002 |
> |  ECBM   | 0.973 $\pm$ 0.001 | 0.713 $\pm$ 0.009 | 0.812 $\pm$ 0.006 | 0.876 $\pm$ 0.000 | 0.478 $\pm$ 0.000 | 0.343 $\pm$ 0.000 | 0.979 $\pm$ 0.000 | 0.854 $\pm$ 0.000 | 0.912 $\pm$ 0.000 |
>
> Your suggestions and guidance have been invaluable in addressing these issues. Thank you for drawing our attention to these points. The new results confirm that our ECBM outperforms these baselines (ProbCBM and PCBM) in both CeleA and AWA2 too.
>
> **Q5. Could you please elaborate more on the specifics of the negative sampling strategy used to learn global $E_{\theta}^{global}$?**
>
> This is a good question. In our current implementation, during each iteration, we randomly select 20% of the samples as negative samples. This strategy has been effective for our experiments, but we acknowledge that more elaborate strategies might be required for larger concept spaces.
>
> In terms of reproducibility and transparency, we commit to providing clear and comprehensive documentation of our method and all related elements upon acceptance of our paper. For the code release, so far, we have finished cleaning up the source code and will release it if the paper is accepted.

---

> ### Author Response · Authors · 2023-11-20
> **[5/5] Thank you for the constructive and encouraging comments.**
>
> **Q6. The outer summation of Preposition 3.2 is very unclear.**
>
> Thank you for mentioning this, and we are sorry for the confusion. In Proposition 3.2, $c$ represents the full vector of concepts and can be broken down into $[c_k, c_{-k}]$, where $c_k$ contain one entry and $c_{-k}$ contains $K-1$ entries. The outer summation is indeed "iterating" over all concept vectors while excluding the $k$-th dimension. We have revised our paper accordingly to clarify this.
>
> **Q7. When learning the positive and negative embeddings for each concept, I was wondering if you have an intuition as to what inductive bias in your model is responsible for forcing the distribution of positive and negative embeddings to be easily separable?**
>
> Our strategy is unique and helps separate positive and negative embeddings. Unlike traditional methods, we enforce the distribution of positive and negative embeddings from the perspective of the input. In our ECBM, we focus on minimizing a specific target.
>
> + When a concept is *active*, we only input the *positive* concept embedding and therefore only the *positive* embedding is trained. During training, minimization of the global energy term $E^{global}(c,y)$ in Eqn. 12 encourages the positive concept embedding to contain more information about the corresponding class label $y$. Similarly, minimization of the concept energy term $E^{concept}(x,c)$ in Eqn. 12 encourages the positive concept embedding to contain more information about the corresponding input $x$.
> + On the other hand, when a concept is *inactive*, we only input the *negative* concept embedding and therefore only the *negative* embedding is trained. During training, minimization of the global energy term $E^{global}(c,y)$ and the concept energy term $E^{concept}(x,c)$ encourages the negative concept embedding to contain the corresponding class label $y$ and input $x$.
> + Empirically we also observed that the positive and negative concept embeddings are sufficiently separable.
>
> In summary, our approach ensures that the distribution of positive and negative embeddings is easily separable, as they are trained under different conditions based on the ground-truth activity of the concept in the training set. Thank you for your insightful question, which has helped us elucidate the unique aspects of our model.
>
>
>
> At the end of this author response, we would like to express our sincere gratitude for your keen eye and attention to detail, which have helped improve the quality of our manuscript. We have addressed the following typos and minor errors you have pointed out. All the changes have been highlighted in blue in our revised version. Your review has been instrumental in enhancing the overall quality of our paper, both in terms of content and presentation. Once again, we appreciate your invaluable input.

---

> > ### Comment · Reviewer_WpSc · 2023-11-22
> >
> > Dear Authors,
> >
> > Thank you so much for all the amazing effort and time you put into answering my (many) questions. I sincerely appreciate all your answers, and I believe that a significant improvement has been made to the manuscript in the new submission. To save space and noise from comments, below I reply to only the leftover/hanging concerns. Nevertheless, do keep in mind that I have read over all your replies and if I don't discuss any of them below, it means that I have found that your reply successfully addressed my question/concern (thank you!).
> >
> > Here are some dangling concerns after your rebuttal:
> >
> > 1. Generally, I appreciate all the new appendices! The biggest complaint I have regarding these appendices is that they are not mentioned in the main body of the paper even when their results are crucial (e.g., Appendix C.5). It may be unrealistic to ask readers to voluntarily dig into the appendix for results that may be crucial for your evaluation (e.g., the `RandInt` results as discussed below). Therefore, I would recommend that all such appendices be moved to the main body when possible or have their key takeaways summarized in the main body of the paper.
> > 2. Regarding error bars: thank you so much for adding these! It is a surprise that so many of them are pretty much 0 (up to three significant figures), particularly for ECBM. Do you have an intuition for this?
> > 2. I would like to point out that error bars are also still missing from the intervention plots/figures.
> > 3. Regarding the lack of `RandInt` in CEM's results for the main paper: I would argue that because `RandInt` is how the CEM authors suggest one should train CEMs and is a key component of their method, when evaluating against CEM one should **always** include `RandInt` to be fair with the author's specified approach. Therefore, I would say it is only fair to report all results that use CEM in this paper based on the CEM trained with `RandInt` (including those in, say, Table 1). If this is not done, then figures like Figure 2 may give the wrong impression that the proposed method is better for interventions than some baselines, when in reality that is not always the case. Also, notice that the lack of `RandInt` is not mentioned anywhere in the updated paper! This means that it is still not clear that the main CEM results in the paper are from a CEM trained without `RandInt` (i.e., not the suggested training approach).
> > 4. Related to the point above, I would suggest including the intervention results across all datasets as part of the main text rather than in the appendix as they show important failure modes that are otherwise not discussed in the main paper.
> > 5. nit: the colors in the legends of figures 12-14 are different from those used for the same methods in figure 11. This makes it a bit confusing to follow. Also, I would suggest using color-blindness-friendly colors and bigger scatter points as they are hard to see at times.
> >
> > Given all these points, I am still a bit hesitant about the effectiveness of this paper's method due to how the method is evaluated. Nevertheless, I believe there is value in these ideas, and I have increased my score towards a *borderline reject* to reflect this after all of the changes made by the authors.  However, I would be willing to increase my score further if the authors address the concerns in this comment before a camera-ready paper is submitted if this paper is accepted. In particular, the issues regarding their evaluation of interventions should be addressed before a new submission is made (e.g., CEMs should be evaluated with `RandInt` and results on all datasets should be included in the main body). This is particularly important as these results currently constitute about a third of the paper's evaluation section yet they fail to be entirely fair and transparent against the baselines they evaluate against (e.g., CEMs).  Similarly, this increase in score comes attached to the promise of the authors releasing code that includes details that would make all of these results easily reproducible for the community.
> >
> > Best of luck with this submission and any future work that may follow it!

---

> > > ### Author Response · Authors · 2023-11-23
> > > **We greatly appreciate your recognition of our efforts and your decision to adjust the score.**
> > >
> > > We greatly appreciate your recognition of our efforts and your decision to adjust the score.
> > >
> > > In response to your concerns, we have revised the evaluation section of the manuscript. Regarding the use of CEMs and RandInt, we have taken your feedback into account and adjusted our approach in the updated version of the paper (**all results are based on CEM with RandInt**).
> > >
> > > All **new changes** are marked in **red** (to distinguish from the first-round changes in blue).
> > >
> > > As for code availability, we assure you that we will make our code open-source and available to the wider research community after acceptance, thereby facilitating the reproducibility of our results.
> > >
> > > **Q1. ...appendices be moved to the main body when possible or have their key takeaways summarized in the main body of the paper.**
> > >
> > > We appreciate your positive feedback regarding the Appendix. In response to your suggestion, we have now relocated all relevant appendices to the main body of the paper when possible. When space is limited, we provided brief summaries of their key takeaways within the main text to ensure essential results are easily accessible to all readers.
> > >
> > >
> > > **Q2. Regarding error bars: thank you so much for adding these! It is a surprise that so many of them are pretty much 0 (up to three significant figures), particularly for ECBM. Do you have an intuition for this?**
> > >
> > > This is a good question.
> > >
> > > + We have noticed a similar phenomenon with ProbCBM in Table 1 of their paper [1], where many error bars are also close to zero.
> > >
> > >
> > > + We believe this could potentially illustrate the stability of our inference phase. The consistently low error range might be indicative of our model's robust performance across multiple runs.
> > >
> > > [1] [Probabilistic Concept Bottleneck Models](https://openreview.net/pdf?id=yOxy3T0d6e) (ICML 2023)
> > >
> > >
> > > **Q3. I would like to point out that error bars are also still missing from the intervention plots/figures.**
> > >
> > > We apologize for the oversight regarding the missing error bars in the intervention plots. We appreciate your attention to detail and have now updated these figures (**Figure 2**) to include error bars. Thank you for pointing this out.
> > >
> > > **Q4. Regarding the lack of RandInt in CEM's results for the main paper:...**
> > >
> > > We agree that to fairly evaluate against CEM, it is essential to include RandInt as it is a key part of the original authors' approach.
> > >
> > > In response to your feedback, we have revised Figure 2 and ensured that all CEM results presented in our paper, including those in Table 1, are based on CEMs trained **with RandInt**. We acknowledge that CEM is a crucial baseline, and we appreciate your guidance on the correct usage of CEM for a fair comparison.
> > >
> > > We have also made it clear in the revised manuscript that the CEM results include RandInt, thereby addressing the lack of clarity in the previous version.
> > >
> > > **Q5. ...including the intervention results across all datasets as part of the main text rather than in the appendix...**
> > >
> > > We appreciate your suggestion to include the intervention results from all datasets in the main body of the paper. Acknowledging their importance, especially in highlighting key failure modes, we have now integrated these results into the main body (**see our new Figure 2**). This should enhance the completeness of our discussion. Thank you for your valuable guidance.
> > >
> > >
> > > **Q6. nit: the colors in the legends of figures...**
> > >
> > > We apologize for the inconsistency in the color schemes between Figures 11 and 12-14. In accordance with your feedback, we have now adjusted the legends to ensure color consistency across all these figures.
> > >
> > > As for your suggestion on using color-blindness-friendly colors and larger scatter points for better visibility, we appreciate your thoughtful advice and have implemented these changes as well.
> > >
> > > Again, we are immensely grateful for your follow-up comments and keeping the communication channel open. While we will not be allowed to see and respond to your further comments after Nov 22 (AOE), we will be sure to incorporate into our revision any further comments you might have after the deadline once these comments are released to us.

---

> > > > ### Comment · Reviewer_WpSc · 2023-11-23
> > > > **Thank you for a fruitful discussion**
> > > >
> > > > Thank you for the quick reply and the very fruitful discussion! I have looked at the updated manuscript, and I believe it has significantly improved since the original submission. To reflect this and the fact that most of my concerns have been addressed by the authors, I have updated my score to a full acceptance.
> > > >
> > > > Best of luck with this work!

---

### Official Review · Reviewer_JBWu · 2023-11-02

**Soundness:** 3 good
**Presentation:** 3 good
**Contribution:** 3 good
**Rating:** 6
**Confidence:** 5

**Summary:**

Concept Bottleneck Models often fail to capture complex interactions between the concepts and the interactions between concepts and labels. The paper proposes ECMBs that define an energy function over input concepts & class for generalizing & addressing the shortcomings of CMBs. Empirical results show that the proposed approach outperforms the CBMs on real-world datasets.

**Strengths:**

Recently, there has been a lot of interest in developing models explaining themselves while making predictions. CBMs were proposed with such criteria in mind.
- The proposed approach addresses some of the shortcomings of CBMs, expanding their availability.
- The paper is well-written and easy to follow.
- Empirical results show that the proposed approach outperforms the CBMs on real-world datasets.

**Weaknesses:**

One of the critical issues with the paper is how they evaluate & the choice of baselines. Even though they consider a diverse set of tasks, the authors must add additional experiments to strengthen the paper.

- Does the architecture of the network affect the performance? Does it need to be shallow or deeper? What are the design considerations for these networks?
- Is the model robust towards background shifts (ref. experiments in [1])
- What is the compute overhead in comparison to CBMs?
- One of the main shortcomings of CBMs is the need for concept annotations. There has been a lot of work on addressing it. Why do the authors build on top of vanilla CBMs, not the improved frameworks [3]?
In addition to CBMs, SENNs also offer a way to develop models that can implicitly explain their predictions, and many of their variants can efficiently encode the relationships between concepts ([2]), as well as between class & concepts. How does the proposed framework compare against those?
- Many of the existing explanation methods aren't robust in practice. Are ECBMs robust to adversarial attacks?

[1] Pang Wei Koh, Thao Nguyen, Yew Siang Tang, Stephen Mussmann, Emma Pierson, Been Kim, Percy Liang: Concept Bottleneck Models. ICML 2020
[2] Anirban Sarkar, Deepak Vijaykeerthy, Anindya Sarkar, Vineeth N. Balasubramanian: A Framework for Learning Ante-hoc Explainable Models via Concepts. CVPR 2022
[3] Tuomas P. Oikarinen, Subhro Das, Lam M. Nguyen, Tsui-Wei Weng: Label-free Concept Bottleneck Models. ICLR 2023

**Questions:**

Refer to Weaknesses

---

> ### Author Response · Authors · 2023-11-20
> **[1/2] Thank you for the constructive and encouraging comments.**
>
> Thank you for your valuable comments. We are glad that you found our model ```"addressing the shortcomings"```, our paper ```"well-written"```/```"easy to follow"```, and our experiments show that our ECBM ```"outperforms the CBMs on real-world datasets"```.
>
>
>
> **Q1. Does the architecture of the network affect the performance? Does it need to be shallow or deeper? What are the design considerations for these networks?**
>
> Indeed, the architecture of the network plays a role in the model's performance. In our case, we've followed the approach implemented by all the baselines, utilizing the ResNet architecture. It is preferable for the network to be deeper. Our experiments show that the fusion of Energy-Based Models (EBM) and Concept Bottleneck Models (CBM) necessitates a deeper network structure; this allows for the extraction of higher-quality feature embeddings, which significantly enhances the model's performance. We also include more details on network architecture in Table 2 of the Appendix.
>
> **Q2. Is the model robust towards background shifts (ref. experiments in [1])**
>
> This is a good question. In response to your comment, we conducted an additional experiment, and the results are included in the table below. These additional results show that on background shift datasets derived from [1], our ECBM achieved a Class accuracy of 0.584 and a Concept accuracy of 0.945. These results are higher than the best Concept Bottleneck Model (CBM) results presented in Table 3 of [1] (i.e., Class accuracy of 0.518 and Concept accuracy of 0.931). We have included the discussion above in our revised paper (e.g., Table 6 of Appendix C) as suggested.
>
> | Methods | Concept | Overall Concept | Class |
> | :--: | :--: | :--: | :--: |
> |Standard | -       | - |   0.373  |
> |Joint (CBM)  | 0.931       | - |   0.518  |
> |Sequential (CBM)   | 0.928       | - |   0.504  |
> |Independent (CBM)   | 0.928      | - |   0.518  |
> |ECBM  | 0.945| 0.416 | 0.584 |
>
> This suggests that our model demonstrates considerable robustness towards background shifts.
>
> [1] Pang Wei Koh, Thao Nguyen, Yew Siang Tang, Stephen Mussmann, Emma Pierson, Been Kim, Percy Liang: Concept Bottleneck Models. ICML 2020
>
> **Q3. What is the compute overhead in comparison to CBMs?**
>
> During the training phase, the computational complexity of our energy-based model (ECBM) is similar to traditional CBMs. During inference, due to the energy-based nature of our model, it requires several iterations, typically between 20 and 100. There are strategies to expedite this inference process. One such approach is to use the predictions from a vanilla CBM as an initialization, thereby 'jump-starting' the optimization and substantially accelerating inference. We believe further research into accelerating energy-based CBMs would be interesting future work. We have included the discussion above in the revision.
>
> **Q4. One of the main shortcomings of CBMs is the need for concept annotations.**
>
> This is a good point. Indeed, the requirement for concept annotations is a known limitation of Concept Bottleneck Models (CBMs).
>
> + We chose to build upon the vanilla CBM as our goal was to delve into the relationships between concepts and between concepts and labels.
> + Moreover, the baseline models we used for comparisons, including PCBM [1], CEM [2], and ProbCBM [3], are also based on vanilla CBM. To ensure a fair comparison, we chose this approach.
> + Our model is compatible with label-free CBMs. Therefore one could potentially use label-free CBMs to generate concept annotations automatically, and train our ECBM using these concepts, thereby addressing this limitation of concept annotation. This way, our ECBM can simultaneously enhance the interpretability, intervention capabilities, and performance of CBM, without manual concept annotations.
>
> Self-Explaining Neural Networks (SENNs) and their variants efficiently encode relationships between concepts (as mentioned in [4]) and between classes and concepts. However, they are not supervised concept bottleneck models and do not allow for direct concept intervention. Therefore they are not applicable to our setting. This is also why previous CBM papers (e.g., our baselines CEM and PCBM) do not include SENNs as baselines. Nonetheless, we acknowledge the importance of these models and have cited and discussed them accordingly in the revision.
>
>
> [1] Post-hoc Concept Bottleneck Models. ICLR, 2023.
>
> [2] Concept Embedding Models. NIPS, 2022.
>
> [3] Probabilistic Concept Bottleneck Models. ICML 2023.
>
> [4] A Framework for Learning Ante-hoc Explainable Models via Concepts. CVPR 2022.

---

> ### Author Response · Authors · 2023-11-20
> **[2/2] Thank you for the constructive and encouraging comments.**
>
> **Q5. Many of the existing explanation methods aren't robust in practice. Are ECBMs robust to adversarial attacks?**
>
> This is a good point. We believe our ECBMs do potentially enjoy stronger robustness to adversarial attacks compared to existing CBM variants. Specifically, our ECBMs are designed to understand the relationships between different concepts, as well as the relationships between concepts and labels. As a result, during inference, ECBMs can leverage these relationships to automatically correct concepts that may be influenced by adversarial attacks. Our preliminary results (e.g., the new experiments on background shifts for **Q2**) suggest that our ECBM can potentially improve the robustness against adversarial attacks compared to existing CBM variants. We believe this is interesting future work; we appreciate your comments and have included the discussion above in the revision.

---

> > ### Comment · Reviewer_JBWu · 2023-11-22
> >
> > Thanks for the detailed response, I will keep my accept recommendation.

---

### Official Review · Reviewer_tSsK · 2023-11-03

**Soundness:** 3 good
**Presentation:** 3 good
**Contribution:** 2 fair
**Rating:** 6
**Confidence:** 4

**Summary:**

The article "Energy-based Concept Bottleneck Models (ECBMs)" presents a novel approach to addressing the limitations of existing concept bottleneck models (CBMs) in providing interpretable insights for black-box deep learning models. The authors identify three key issues with conventional CBMs, namely their inability to capture high-order interactions between concepts, quantify complex conditional dependencies, and strike a balance between interpretability and performance.

The proposed ECBMs leverage neural networks to define a joint energy function for input, concept, and class label tuples. This unified interface allows for the representation of class label prediction, concept correction, and conditional interpretation as conditional probabilities, enabling a more comprehensive understanding of the model's behavior.

**Strengths:**

1. To my knowledge, utilizing energy based models in the context of CBMs is novel. To my knowledge, this is the first paper that does so.
2. The writing is good.
3. The experiments are pretty comprehensive but i have some points

**Weaknesses:**

One caveat is I am not that familiar with energy based models, so I wont be able to comment on the inherent problems of energy based models. However, given the context of CBMs, i want to have the following criticism:

1. The second gap: Given the class label and one concept, the probability of predicting another concept? is not clear
2. Incomplete literature review. The authors should add the following papers as each of them has a unique perspective to solve the problem.

[1] Interpretable Neural-Symbolic Concept Reasoning. Barberio et al. ICML 2023

[2] Entropy-based Logic Explanations of Neural Networks. Barberio et al. AAAI 2021

[3] Addressing Leakage in Concept Bottleneck Models. Havasi et. al., Neurips 2022

[4] Dividing and Conquering a BlackBox to a Mixture of Interpretable Models: Route, Interpret, Repeat. Ghosh et al. ICML 2023

[5] Distilling BlackBox to Interpretable models for Efficient Transfer Learning. Ghosh et al. MICCAI 2023

[6] Language in a Bottle: Language Model Guided Concept Bottlenecks for Interpretable Image Classification .Yang et al. CVPR 2023

[8] A Framework for Learning Ante-hoc Explainable Models via Concepts Sarkar et al. CVPR 2022

[9] Label-Free Concept Bottleneck Models. Oikarinen et al. ICLR 2023

3. How does each of three conditional probabilities - p(y|x), p(c-k|x, ck), and p(ck|y, ck′)—mitigate the 3 limitations - Interpretability, Intervention, Performance?
4. How does this method impact label free CBM or Language in a bottle kind of framework? There concepts are retrieved from the language model. because in more realistic setting like medical images having ground truth concepts is expensive.
5. I believe this is an inherently interpretable method. How to integrate it in a posthoc setting like PCBM? Also, if it can be integrated in a posthoc setting, i would like to see the comparison of completeness scores.
6. There is a new notion of difficulty estimation for samples using concept bottleneck models in . Can the energy based models be extended to that?
7. One limitation of concept bottleneck models is the limited applicability in a real-life scenario. I would like to see one experiment on an application - for example shortcut identification as in [4] and Discover and Cure: Concept-aware Mitigation of Spurious Correlation. Wu et al. ICML 2023
8. Also, there is a problem of leakage in CBM as in [3]. Will the ECBM be helpful in that regard? If the authors can show ECBM is better for the experiments depicted in fig. 2 and 3 in [3], this paper will be very strong.


Overall, the contribuition of energy based method for CBM is novel. However, the x-ai community has identified the two setbacks of concept bottleneck models - 1) the cost of getting the ground truth concepts (for inherently interpretable setup) 2) the concepts are purely correlation not causal (for posthoc setup), so it is not guranteed if the concepts are actually utilized by the blackbox. This paper does not answer this two core research gap through their method as how ECBMs can help in either of the two question is still not clear to me.

**Extremely pleased with response from the authors and especially with the leakage experiments. This paper deserves a 7 for sure and 8 is too high. So i am increasing my score**

**Questions:**

See weakness

---

> ### Author Response · Authors · 2023-11-20
> **[1/2] Thank you for the constructive and encouraging comments.**
>
> We value the feedback you have provided. We are glad that you found the problem we solve ```"novel"```, our presentation ```"good"```, and our experiments ```"comprehensive"```.  We will address each of your comments in turn below.
>
>
> **Q1.  The second gap: Given the class label and one concept, the probability of predicting another concept? is not clear**
>
> We appreciate your question about the interpretation of the second gap. This scenario is an example of conditional interpretation in our model. In this context, given a class label $y$ and one concept $c$, the model predicts the probability of another concept $c'$. This process provides insights such as: "from the model's perspective, how relevant concepts $c$ and $c'$ are for images in class $y$".
>
> We understand that this explanation might not fully address your comment. If you could provide further details or specify the aspects that are unclear, we would be very glad to offer a more targeted explanation.
>
> **Q2. Incomplete literature review.**
>
> We appreciate your feedback about the comprehensiveness of our literature review. Your suggestions have indeed enhanced the breadth of our research and have been incorporated into the 'Related Work' section of our revised paper.
>
> **Q3. How does each of three conditional probabilities - p(y|x), p(c-k|x, ck), and p(ck|y, ck′)—mitigate the 3 limitations - Interpretability, Intervention, Performance?**
>
> Each of the three conditional probabilities addresses different limitations related to Interpretability, Intervention, and Performance:
>
> - $p(c, y|x)$ contributes to **performance** improvment. Unlike original CBMs, which predict concepts first and then predict labels using these predicted concepts, our model predicts concepts $c$ and class labels $y$ *simultaneously* while considering their *compatibility*. Our global energy network measures the dependency between $c$ and $y$, thereby improving the prediction accuracy.
> - $p(c_{-k}|x,c_k)$ is designed for **intervention**. For instance, given the input $x$, if an expert knows the correct concept $c_k$ and performs concept intervention on concept $k$, this can help correct other concepts $c_{-k}$. This is equivalent to computing the conditional probability $p(c_{-k}, y|x,c_k)$.
> - $p(c|y)$ and $p(c_{k}|y,c_{k^{\prime}})$ are used for **interpretability**. $p(c_{k}|y,c_{k^{\prime}})$ indicates the strength of the connection between $c_k$ and $c_{k^{\prime}}$ for a certain class $y$ of images, providing conditional interpretation. $p(c|y)$ highlights which concepts are important to the given class.
>
> We hope that these explanations provide clarity on your question, and we are open to further discussions on this.
>
> **Q4. How does this method impact label free CBM or Language in a bottle kind of framework?**
>
> This is a good question. Indeed, our ECBM is compatible with the label-free CBM method. One possible approach is to initially extract concepts using the label-free method and subsequently train our ECBM. This would be interesting future work. We appreciate your valuable insights and have incorporated this comment into the revision.
>
> **Q5.  I believe this is an inherently interpretable method. How to integrate it in a posthoc setting like PCBM?**
>
> Yes, ECBM is an inherently interpretable method, not a post-hoc one. In this paper, we concentrate on inherent interpretability. Although integrating our method into a post hoc setting like PCBM is outside the scope of this study, we agree that it would be interesting and nontrivial future work to explore. We are grateful that you have accurately grasped the nature of our method, and will make this point clearer in the revision as suggested.
>
> **Q6. There is a new notion of difficulty estimation for samples using concept bottleneck models in .**
>
> We apologize for any confusion caused, but there seems to be a typo in your question. We are uncertain about the specific paper in which the notion of difficulty estimation is discussed. We examined the nine papers you provided, but could not locate this term. If you could clarify or provide more information regarding this, we would greatly appreciate it. Thank you for your understanding.

---

> ### Author Response · Authors · 2023-11-20
> **[2/2] Thank you for the constructive and encouraging comments.**
>
> **Q7. One limitation of concept bottleneck models is the limited applicability in a real-life scenario...shortcut identification...**
>
> Thank you for pointing us to these interesting papers. We have carefully gone through them and find that our model is actually equipped to carry out such tasks.
>
>
> Specifically, $p(c|y)$ generated by our ECBM shows the importance of each concept for each class. By comparing our ECBM's estimated $p(c|y)$ with the ground-truth $p(c|y)$, we can identify any significant differences in concept importance to shed light on any spurious feature learned in the model. For example, if our ECBM estimates that $p(c_1=1|y=1) = 0.8$ while the groud-truth $p(c_1=1|y=1) = 0$, this indicates that the model learns concept $c_1$ as a spurious feature.
>
> Our experiments show that our ECBM is robust against spurious correlations. For example, Figure 3 in the paper shows that our ECBM's estimated $p(c|y)$ is very close to the ground truth. This is possibly due to our ECBM's joint inference of class labels and concepts, which helps mitigate spurious correlation. Furthermore, we conducted additional experiments on the TravelingBirds dataset following the robustness experiments of CBM concerning background shifts. The results (Appendix C's Table 6) reveal that our ECBM outperforms CBMs in this regard. These findings underscore our model’s superior robustness to spurious correlations.
>
> We have included the discussion and the references in our revision as suggested.
>
>
> **Q8. Also, there is a problem of leakage in CBM as in [3]. Will the ECBM be helpful in that regard?**
>
> This is an insightful question, and thanks for pointing us to [3]. We believe that our ECBM model could indeed offer a solution to this problem. In response to your suggestion, we conducted an experiment similar to the ones depicted in Figures 2 and 3 in [3], but using our ECBM model. We found that when only a few concepts were present, the accuracy of our method did not significantly increase. However, there was a noticeable increase in the performance of the baseline model(like CEM) at this stage. Our model's accuracy gradually increased when more concepts were given during training. This suggests that *our ECBM model can help mitigate the leakage problem*; in contrast, baselines such as CBM and CEM still suffer from some degree of leakage due to their soft concepts. We have included these additional empirical results in Figure 11 of Appendix C.5 of our revised manuscript. Thank you for your valuable suggestion, which has helped us strengthen our evaluation of ECBM.
>
> [3] Addressing Leakage in Concept Bottleneck Models. Havasi et. al., Neurips 2022.
>
> **Q9. Two setbacks of concept bottleneck models.**
>
> Your question highlights some significant concerns prevalent among the CBM community.
>
> **(1) The cost of obtaining ground truth concepts:** This issue can be mitigated using label-free CBM. As mentioned in response to Q4, our model is compatible with label-free CBM. We can employ label-free methods to generate the concepts, which can then be used to train our ECBM.
>
> **(2) The non-causal correlation of concepts:** Our paper focuses on inherently interpretable models rather than the post hoc setup. Therefore we do not claim that ECBM learns causal relation. The goal of our ECBM is to learn a concept-based model that
>
> + uses different *conditional* (*not causal*) probabilities to provide a comprehensive concept-based interpretation of the prediction model and
> + improve prediction accuracy by considering compatibility among different concepts and class labels.
>   It is worth noting that the vanilla CBM also differs from causal settings, as it does not require concepts to directly cause labels but instead requires that concepts are highly predictive of labels.
>
> We hope this addresses your concerns and provides clarity on how ECBMs can potentially bridge these research gaps.

---

> ### Author Response · Authors · 2023-11-23
> **Thank you for your time and effort in reviewing our paper**
>
> Dear Reviewer tSsK,
>
> Thank you for your time and effort in reviewing our paper.
>
> We firmly believe that our response and revisions can fully address your concerns. We are open to discussion (before Nov 22 AOE, after which we will not be able to respond to your comments unfortunately) if you have any additional questions or concerns, and if not, we will be immensely grateful if you could reevaluate your score.
>
> Thank you again for your reviews which helped to improve our paper!
>
> Best regards,
>
> ECBM Authors

---

### Official Review · Reviewer_XFNY · 2023-11-06

**Soundness:** 2 fair
**Presentation:** 3 good
**Contribution:** 2 fair
**Rating:** 6
**Confidence:** 4

**Summary:**

This paper has introduced the idea of enery-based models in the concept bottleneck models to capture the interaction between the so-called "concepts" and quantify the dependencies between the "concepts" and class labels.

**Strengths:**

The paper enhances the state-of-the-art Concept Bottleneck Models by introducing a unified energy formulation that integrates concept-based prediction, concept correction, and conditional interpretation into a cohesive framework.

**Weaknesses:**

1. I find it difficult to fully embrace the idea of Concept Bottleneck Modeling (CBM) presented in this paper, as well as in prior research, due to the absence of a clear definition for the term "concepts." From my perspective, what is referred to as "concepts" in this context appears to be nothing more than additional labels or explicit features that can be readily discerned by humans. However, one of the notable advantages of deep learning networks lies in their ability to automatically extract features, even those that may not be easily identifiable by humans. Consequently, I believe that the primary reason for the observed decline in performance of CBMs, when compared to recent work such as [1] on CUB, is that the model is exclusively directed towards learning these so-called "concepts" and neglects the acquisition of features that can be explicitly identified.
2. There missing baseline models that are not based on CBM in the experiments.
3. There missing ablation study on the proposed method with only a single branch, i.e, the 'concepts' branch and the 'class' branch as shown in Figure 1.
4. The "related work" section about energy-based models is not complete. It misses some pioneering works and representative works to build a full history of the advancement of EBMs. For example, [2] [3] are pioneering EBM works, [4] also use an EBM for the distribution of labels (or other information) and data, which also captures high-level concepts and data. It is better to have a comparative discussion on both models if they share a similar concept.

[1] Progressive co-attention network for fine-grained visual classification. 2021 International Conference on Visual Communications and Image Processing (VCIP). IEEE, 2021.

[2] A Theory of Generative ConvNet. ICML 2016

[3] Cooperative Training of Descriptor and Generator Networks. TPAMI 2018.

[4] Cooperative Training of Fast Thinking Initializer and Slow Thinking Solver for Conditional Learning. TPAMI 2021.

**Questions:**

1. As demonstrated in Table 1, the gap in performance between CBM and ECBM is noticeably smaller on the AWA2 dataset compared to that observed on the CUB and CelebA datasets. Is there any explanation for this phenomenon?
2. How can the model ensure a stable solution during inference while searching for the optimal values of 'c' and 'y'? Is there any complexity analysis in B.1.
3. Is it possible for the model to detect undefined 'concepts,' or is it capable of exploring new 'concepts'?

---

> ### Author Response · Authors · 2023-11-20
> **[1/4] Thank you for the constructive and encouraging comments.**
>
> Thank you for your valuable reviews. We are glad that you found our model ```"enhances the state-of-the-art"```. Below we address your questions one by one:
>
> **W1.  "... due to the absence of a clear definition of the term "concepts" ... neglects the acquisition of features that can be explicitly identified."**
>
> This is a good question.
>
> **The Advantages of Concept Bottleneck Models (CBMs):** We agree that "concepts" can be seen as "explicit features that can be readily discerned by humans". However, we would like to emphasize that by using these concepts as the bottleneck, Concept Bottleneck Models (CBMs) and their variants have additional advantages:
>
> + Humans can intervene in these predicted concepts, thereby influencing final predictions. This is particularly beneficial in real-world settings, where it is crucial not only to provide interpretability but also to incorporate human participation into the model's decision-making process, known as the human-in-the-loop process.
>
> + They predict interpretations that can be presented to humans for verification, creating a feedback loop for accuracy.
>
>
> **Addressing Performance Decline:** Thanks for pointing us to the interesting paper [1], which we have cited and discussed in the revision. We agree that maintaining concept-based interpretability while ensuring model performance is an open question in the field. The baselines we compare in our work all strive to address this issue.
>
> Our proposed ECBM further pushes the limit by utilizing an energy network. Specifically, by enabling the learning of compatibility between input $x$, concept $c$, and class label $y$, we aim to create a network that can capture features that are not readily identifiable. Looking at the empirical results (refer to Appendix C, Table 4), our ECBM achieves performance very close to the black-box model on the CUB and AWA2 datasets; for example, on CUB, the accuracy of the black-box model and our ECBM is $0.826$ and $0.812$, respectively. In some cases, such as with the CelebA dataset, it even improves upon the black-box performance ($0.343$ versus $0.291$). This can be attributed to our joint inference process of concepts and class labels.
>
> [1] Progressive co-attention network for fine-grained visual classification. 2021 International Conference on Visual Communications and Image Processing (VCIP). IEEE, 2021.

---

> ### Author Response · Authors · 2023-11-20
> **[2/4] Thank you for the constructive and encouraging comments.**
>
> **W2. "...missing baseline models that are not based on CBM in the experiments."**
>
> Thank you for mentioning this. Our current focus is primarily on concept-based models and, as such, our comparisons are chiefly made with standard CBMs, with an emphasis on both interpretability and model performance.
> In line with similar studies, such as those on PCBM [1] and CEM [2], we've chosen to include only CBM variants as baselines. This decision is based on the fact that non-CBM models do not possess both concept interpretability and direct concept intervention capabilities, which are crucial for our line of investigation.
>
> Nevertheless, we understand the potential value of incorporating traditional black-box models as baselines to provide a more comprehensive perspective. In addition to our primary findings, we have conducted tests on a black-box model with the same network architecture. Please kindly find these results below and in Table 4 of Appendix C in our revised paper. These results show that
>
> + our ECBM achieves performance very close to the black-box model on the CUB and AWA2 datasets; for example, on CUB, the accuracy of the black-box model and our ECBM is $0.826$ and $0.812$, respectively.
> + In some cases, such as with the CelebA dataset, it even improves upon the black-box performance ($0.343$ versus $0.291$). This can be attributed to our joint inference process of concepts and class labels.
> + our ECBM outperforms all baseline models, including CBM and CEM.
>
>
>
> | Model/Dataset |         |       CUB       |       |         |     CelebA      |       |         |      AWA2       |       |
> | :------------: | :-----: | :-------------: | :---: | :-----: | :-------------: | :---: | :-----: | :-------------: | :---: |
> |     Metric     | Concept | Overall Concept | Class | Concept | Overall Concept | Class | Concept | Overall Concept | Class |
> |    **ECBM**    |  0.973  |      0.713      | 0.812 |  0.876  |      0.478      | 0.343 |  0.979  |      0.854      | 0.912 |
> |      CBM       |  0.964  |      0.364      | 0.759 |  0.837  |      0.381      | 0.246 |  0.979  |      0.803      | 0.907 |
> |      CEM       |  0.965  |      0.396      | 0.796 |  0.876  |      0.457      | 0.330 |  0.978  |      0.796      | 0.908 |
> |    Blackbox    |    -    |        -        | 0.826 |    -    |        -        | 0.291 |    -    |        -        | 0.929 |
>
> If you have any specific suggestions for such baselines, please do not hesitate to share them. We would be more than willing to conduct additional comparative experiments accordingly.
>
> [1] Post-hoc Concept Bottleneck Models. ICLR, 2023.
>
> [2] Concept Embedding Models. NIPS, 2022.

---

> ### Author Response · Authors · 2023-11-20
> **[3/4] Thank you for the constructive and encouraging comments.**
>
> **W3. "There missing ablation study on the proposed method with only a single branch, i.e, the 'concepts' branch and the 'class' branch as shown in Figure 1."**
>
> Thank you for your astute observation regarding the absence of an ablation study focused on a single branch method. In response to your comment, we have conducted an additional ablation study and incorporated the results into our work.
>
> Specifically, we include two baselines for this ablation study:
>
> + x-y: This baseline involves a single branch that utilizes the class energy network to directly predict the class label.
>
> + x-c-y: This baseline, on the other hand, employs a single branch that first uses the concept energy network to predict the concepts. Following this, it uses the global energy network to predict the class label.
>
> The findings are as follows:
>
> | Branch/Dataset |         |       CUB       |       |         |     CelebA      |       |         |      AWA2       |       |
> | :------------: | :-----: | :-------------: | :---: | :-----: | :-------------: | :---: | :-----: | :-------------: | :---: |
> |     Metric     | Concept | Overall Concept | Class | Concept | Overall Concept | Class | Concept | Overall Concept | Class |
> |      x-y       |    -    |        -        | 0.825 |    -    |        -        | 0.265 |    -    |        -        | 0.909 |
> |     x-c-y      |  0.968  |      0.680      | 0.726 |  0.870  |      0.464      | 0.175 |  0.979  |      0.864      | 0.905 |
> |    **ECBM**    |  0.973  |      0.713      | 0.812 |  0.876  |      0.478      | 0.343 |  0.979  |      0.854      | 0.912 |
> |      CBM       |  0.964  |      0.364      | 0.759 |  0.837  |      0.381      | 0.246 |  0.979  |      0.803      | 0.907 |
> |      CEM       |  0.965  |      0.396      | 0.796 |  0.876  |      0.457      | 0.330 |  0.978  |      0.796      | 0.908 |
>
>
> We have also included this table detailing these results in Table 4 of Appendix C for further reference.
>
> From this new ablation study, we have the following observations:
>
> + Our full model demonstrates superior performance when tested on the CelebA and AWA2 datasets, outperforming all other baseline models, and verifying the effectiveness of each component of our ECBM.
>
> + For the CUB dataset, our ECBM $x-y$ single branch surprisingly approaches the performance of the black-box model. However, it lacks concept interpretability. In contrast, our full ECBM can provide such concept interpretability, albeit with a slight decrease in class accuracy. Note that our full ECBM still outperforms all baseline models, including "x-y", "x-c-y", CBM, and CEM, verifying the effectiveness of each component of our ECBM.
>
> **W4. The "related work" section about energy-based models is not complete.**
>
> We appreciate your insightful feedback on the 'Related Work' section of our paper. We agree that a more comprehensive review of the historical development and pioneering works of Energy-Based Models (EBMs) would enhance our discussion. We recognize the importance of the works you have cited, [2], [3], and [4], in the field of EBMs. Particularly, the use of an EBM for the distribution of labels (or other information) and data in [4] aligns closely with our research as it also captures high-level concepts and data. In light of your suggestions, we have updated our paper to include these seminal works.
>
> [2] A Theory of Generative ConvNet. ICML 2016
>
> [3] Cooperative Training of Descriptor and Generator Networks. TPAMI 2018.
>
> [4] Cooperative Training of Fast Thinking Initializer and Slow Thinking Solver for Conditional Learning. TPAMI 2021.
>
> ## For Questions:
>
> **Q1. As demonstrated in Table 1, the gap in performance between CBM and ECBM is noticeably smaller on the AWA2 dataset compared to that observed on the CUB and CelebA datasets. Is there any explanation for this phenomenon?**
>
> This is a good question. As indicated in Table 1, the CBM already achieves high performance on the AWA2 dataset, with Concept accuracy at 0.979, Overall concept accuracy at 0.803, and Class accuracy at 0.907. Given such high accuracy, there is very limited room for further improvement. It is also worth noting that other baseline models, such as ProbCBM and CEM, do not even surpass the performance of the original CBM on the AWA2 dataset. Despite the narrower performance gap, our ECBM model still achieves the state-of-the-art results on this dataset, underscoring its effectiveness.

---

> > ### Comment · Reviewer_XFNY · 2023-11-22
> >
> > According to the results of x-y and x-c-y in the table, could we conclude that the performance drops with the introduction of concepts or interpretation ability?

---

> > > ### Author Response · Authors · 2023-11-22
> > > **Thank you for your follow-up comments and for keeping the communication channel open.**
> > >
> > > Thank you for the follow-up comments and for keeping the communication channel open.
> > >
> > > This is a good question. In fact, comparing the results of x-y and x-c-y in the table *cannot* conclude that ``"the performance drops with the introduction of concepts or interpretation ability"``. Below are the reasons:
> > >
> > > - x-c-y is an **ablated** version of ECBM, not our full ECBM. x-c-y first predicts concepts $c$ through the "concept" branch and then predicts class labels $y$ using the predicted concepts.
> > > - The fact that x-c-y performs worse than x-y means that the performance drops if one introduces concepts **naively**. The goal of our **full ECBM** is exactly to address such performance drop.
> > > - In fact, if we compare the performance of the full ECBM and x-c-y, we can see that the full ECBM outperforms x-c-y in all three datasets in terms of class accuracy. This demonstrates the effectiveness of our **full ECBM**'s joint training and inference algorithms.
> > > - Note that our **full ECBM** actually outperforms x-y on both *CelebA* and *AWA2*, particularly on CelebA, where the **full ECBM**'s accuracy (0.343) significantly exceeds that of **x-y** (0.265). This shows that the performance actually **increases** with the **sophisticated** introduction (as opposed to the **naive** version of x-y) of concepts or interpretation ability.

---

> ### Author Response · Authors · 2023-11-20
> **[4/4] Thank you for the constructive and encouraging comments.**
>
> **Q2. How can the model ensure a stable solution during inference while searching for the optimal values of 'c' and 'y'? Is there any complexity analysis in B.1.**
>
> As delineated in Appendix B, Algorithm 1, the computational complexity of our proposed method is $O(TN)$, where $T$ denotes the number of iterations, and $N$ denotes the number of parameters in the neural networks. Thus, the complexity scales linearly with both the number of iterations and parameters.
>
> Empirically, we have observed that the inference process is generally stable. For the gradient-based inference process, we employed the Adam optimizer, which has shown consistent and reliable performance.
>
> As shown in the updated Table 1 in the main paper, we ran our ECBM with 5 different random seeds and report the means and standard deviations. Across all datasets and all metrics, our standard deviations range from 0.000 to 0.006, except for the overall concept accuracy for CUB, which is 0.009. All values are represented up to three decimal digits. These results confirm the stability of our ECBMs.
>
>
> **Q3. Is it possible for the model to detect undefined 'concepts,' or is it capable of exploring new 'concepts'?**
>
> Thank you for this insightful question. Indeed, our model does have the potential to explore and identify new 'concepts'. Specifically, we can "reserve" additional "undefined" concept embeddings to allow the model to learn them, and introduce an inductive bias to ensure these embeddings are as distinct as possible from each other and from existing concepts. We believe this capability for concept discovery presents a promising direction for future research. In response to your question, we have incorporated a discussion on this aspect into the revision.
>
> We hope this response addresses your concerns adequately. We appreciate your insightful comments and look forward to further discussions on improving our work.

---

> ### Author Response · Authors · 2023-11-23
> **Thank you for your time and effort in reviewing our paper**
>
> Dear Reviewer XFNY,
>
> Thank you for your time and effort in reviewing our paper.
>
> We firmly believe that our response and revisions can fully address your concerns. We are open to discussion (before Nov 22 AOE, after which we will not be able to respond to your comments unfortunately) if you have any additional questions or concerns, and if not, we will be immensely grateful if you could reevaluate your score.
>
> Thank you again for your reviews which helped to improve our paper!
>
> Best regards,
>
> ECBM Authors

---

### Author Response · Authors · 2023-11-20
**Response to all the reviewers and area chairs.**

We thank all reviewers for their valuable comments. We are glad that they found the problem we sovle ```"novel"```/```"providing interpretable insights"```/```"significant"```/```"excellent"```(tSsK, WpSc, doCC), our model ```"enhances the state-of-the-art"```/```"cohesive"```/```"addressing the shortcomings"```/```"original and interesting"```(XFNY, JBWu, WpSc, doCC), our paper ```"good"```/```"well-written"```/```"easy to follow"```/```"concise and clear"```(tSsK, JBWu, WpSc), and our experiments ```"comprehensive"``` (tSsK) and show that our ECBM ```"outperforms the CBMs on real-world datasets"```(JBWu).

Below we address the reviewers’ questions. We have also updated both the main paper and the Appendix (with the changed part marked in blue).

---

### Meta-Review · Area_Chair_kz8D · 2023-12-05

**Metareview:**

The paper introduces an energy-based concept bottleneck model (ECMB) as a solution to the limitations of Concept Bottleneck Models (CMBs), which often struggle to capture intricate interactions among concepts and between concepts and labels. The proposed ECMB defines an energy function over input concepts and classes, aiming to enhance generalization. Empirical results indicate superior performance on real-world datasets compared to traditional CBMs. The paper is well-organized and well-written, presenting a well-motivated and novel framework. The rebuttal adequately addresses the majority of concerns raised by the reviewers, leading to a consensus favoring acceptance of the paper. The AC aligns with the reviewers' opinions and recommends accepting the paper. The AC encourages the authors to incorporate the valuable suggestions provided by the reviewers in their revision.

**Justification For Why Not Higher Score:**

The current paper is considered borderline for acceptance. While the contribution may not be exceptionally significant, it could still be of interest to the machine learning community. Therefore, the Area Chair recommends accepting the paper as a poster.

**Justification For Why Not Lower Score:**

The paper is well-written, and its idea is interesting and new. All reviewers unanimously agree to accept the paper, as the rebuttal effectively addresses the concerns they raised.

---

### Decision · Program_Chairs · 2024-01-16

Accept (poster)